# Rank Diminishing in Deep Neural Networks

**Ruili Feng[1], Kecheng Zheng[2,1], Yukun Huang[1], Deli Zhao[2,3],**
**Michael Jordan[4], Zheng-Jun Zha[1]***
[1]University of Science and Technology of China, Hefei, China
[2]Ant Research, [3]Alibaba Group, Hangzhou, China
[4]University of California, Berkeley
`ruilifengustc@gmail.com,{zkcys001,kevinh}@mail.ustc.edu.cn,`
`zhaodeli@gmail.com,jordan@cs.berkeley.edu,zhazj@ustc.edu.cn.`

## Abstract

The rank of neural networks measures information flowing across layers. It is
an instance of a key structural condition that applies across broad domains of
machine learning. In particular, the assumption of low-rank feature representations
leads to algorithmic developments in many architectures. For neural networks,
however, the intrinsic mechanism that yields low-rank structures remains vague and
unclear. To fill this gap, we perform a rigorous study on the behavior of network
rank, focusing particularly on the notion of rank deficiency. We theoretically
establish a universal monotonic decreasing property of network rank from the basic
rules of differential and algebraic composition, and uncover rank deficiency of
network blocks and deep function couplings. By virtue of our numerical tools,
we provide the first empirical analysis of the per-layer behavior of network rank
in practical settings, *i.e.*, ResNets, deep MLPs, and Transformers on ImageNet.
These empirical results are in direct accord with our theory. Furthermore, we reveal
a novel phenomenon of independence deficit caused by the rank deficiency of
deep networks, where classification confidence of a given category can be linearly
decided by the confidence of a handful of other categories. The theoretical results
of this work, together with the empirical findings, may advance understanding of
the inherent principles of deep neural networks. Code to detect the rank behavior
of networks can be found in https://github.com/RuiLiFeng/Rank-Diminishing-in-
Deep-Neural-Networks.

## 1 Introduction

In mathematics, the rank of a smooth function measures the volume of independent information
captured by the function [21]. Deep neural networks are highly smooth functions, thus the rank
of a network has long been an essential concept in machine learning that underlies many tasks
such as information compression [46, 55, 35, 53, 47], network pruning [31, 54, 6, 25, 10], data
mining [7, 24, 11, 56, 19, 28], computer vision [58, 57, 30, 26, 28, 60, 59, 16, 17], and natural
language processing [9, 27, 8, 12]. Numerous methods are either designed to utilize the mathematical
property of network ranks, or are derived from an assumption that low-rank structures are to be
preferred.

Yet a rigorous investigation to the behavior of rank of general networks, combining both theoretical
and empirical arguments, is still absent in current research, weakening our confidence in the being able
to predict performance. To the best of our knowledge, there are only a few previous works discussing
the rank behavior of specific network architectures, like attention blocks [15] and BatchNorms [13, 5]

---

*Corresponding author.

36th Conference on Neural Information Processing Systems (NeurIPS 2022).

in pure MLP structures. The empirical validation of those methods are also limited to shallow networks, specific architectures, or merely the final layers of deep networks, leaving the global behavior of general deep neural networks mysterious due to prohibitive space-time complexity for measuring them. Rigorous work on network rank that combines both strong theoretical and empirical evidence would have significant implications.

In this paper, we make several contributions towards this challenging goal. We find that the two essential ingredients of deep learning, the chain rules of differential operators and matrix multiplications, are enough to establish a universal principle—that network rank decreases monotonically with the depth of networks. Two factors further enhance the speed of decreasing: a) the explicit rank deficiency of many frequently used network modules, and b) an intrinsic potential of spectrum centralization enforced by the nature of coupling of massive composite functions. To empirically validate our theory, we design numerical tools to efficiently and economically examine the rank behavior of deep neural networks. This is a non-trivial task, as rank is very sensitive to noise and perturbation, and computing ranks of large networks is computationally prohibitive in time and space. Finally, we uncover an interesting phenomenon of independence deficit in multi-class classification networks. We find that many classes do not have their own unique representations in the classification network, and some highly irrelevant classes can decide the outputs of others. This independence deficit can significantly deteriorate the performance of networks in generalized data domains where each class demands a unique representation. In conclusion, the results of this work, together with the numerical tools we invent, may advance understanding of intrinsic properties of deep neural networks, and provide foundations for a broad study of low-dimensional structures in machine learning.

## 2 Preliminaries

**Settings**    We consider the general deep neural network with $L$ layers. It is a smooth vector-valued function $\boldsymbol{F} : \mathbb{R}^n \to \mathbb{R}^d$, where $\mathbb{R}^n$ and $\mathbb{R}^d$ are the ambient space of inputs and outputs, respectively. Deep neural networks are coupling of multiple layers, thus we write $\boldsymbol{F}$ as

$$\boldsymbol{F} = \boldsymbol{f}^L \circ \boldsymbol{f}^{L-1} \circ \cdots \circ \boldsymbol{f}^1. \tag{1}$$

For simplicity, we further write the $k$-th sub-network[2] of $\boldsymbol{F}$ as

$$\boldsymbol{F}_k = \boldsymbol{f}^k \circ \cdots \circ \boldsymbol{f}^1, \tag{2}$$

and we use $\mathcal{F}_k = \boldsymbol{F}_k(\mathcal{X})$ to denote the feature space of the $k$-th sub-network on the data domain $\mathcal{X}$. We are more interested in the behavior of network rank in the feature spaces rather than scalar outputs (which trivially have rank 1). Thus, for classification or regression networks that output a scalar value, we will consider $\boldsymbol{F} = \boldsymbol{F}_L$ as the transformation from the input space to the final feature space instead. Thus, we always have $n \gg 1$ and $d \gg 1$. For example, for ResNet-50 [20] architecture on ImageNet, we only consider the network slice from the inputs to the last feature layer with 2,048 units.

**Rank of Function**    The rank of a function $\boldsymbol{f} = (\boldsymbol{f}_1, ..., \boldsymbol{f}_d)^T : \mathbb{R}^n \to \mathbb{R}^d$ refers to the highest rank of its Jacobian matrix $J_{\boldsymbol{f}}$ over its input domain $\mathcal{X}$, which is defined as

$$\mathrm{Rank}(\boldsymbol{f}) = \mathrm{Rank}(\boldsymbol{J}_{\boldsymbol{f}}) = \max_{\boldsymbol{x} \in \mathcal{X}} \mathrm{Rank}\left((\partial \boldsymbol{f}_i(\boldsymbol{x})/\partial \boldsymbol{x}_j)_{n \times d}\right). \tag{3}$$

It is well-known that the region of non-highest rank is a zero-measure set on the feature manifold by Sard's theorem [21], so we are safe to ignore them in the definition of the function ranks. The rank of a function represents the volume of information captured by it in the output [21]. That is why it is so important to investigate the behavior of neural networks and many practical applications. Theoretically, by the rank theorem and Sard's theorem of manifolds [21], we can know that rank of the function equals the intrinsic dimension of its output feature space, as in the following lemma.[3]

**Lemma 1.** *Suppose that $\boldsymbol{f} : \mathbb{R}^n \to \mathbb{R}^d$ is smooth almost everywhere. Let $\mathrm{Rank}(\boldsymbol{f}) = r$. If data domain $\mathcal{X}$ is a manifold embedded in $\mathbb{R}^n$ and $\boldsymbol{\phi} : \mathcal{U} \to \mathcal{O}$ is a smooth bijective parameterization from an open subset $\mathcal{U} \subset \mathbb{R}^s$ to $\mathcal{O} \subset \mathcal{X}$, then we have $\dim(\boldsymbol{f}(\mathcal{X})) = \mathrm{Rank}(\boldsymbol{J}_{\boldsymbol{f} \circ \boldsymbol{\phi}}) \leq r$. Thus, the rank of function $\boldsymbol{f}$ gives an upper bound for the intrinsic dimension $\dim(\boldsymbol{f}(\mathcal{X}))$ of the output space.*

---

[2]In this paper, sub-network means network slice from the input to some intermediate feature layer; layer network means an independent component of the network, without skip connections from the outside to it, like bottleneck layer of ResNet-50.

[3]Due to space limitation, all the related proofs are attached in the Appendix.

It is worth mentioning that the intrinsic dimension $\dim(\boldsymbol{f}(\mathcal{X}))$ of the feature space is usually hard to measure, so the rank of the network gives an operational estimate of it.

# 3 Numerical Tools

Validating the rank behavior of deep neural networks is a challenging task because it involves operations of high complexity on large-scale non-sparse matrices, which is infeasible both in time and space. Computing the full Jacobian representation of sub-networks of ResNet-50, for example, consumes over 150G GPU memory and several days at a single input point. In accuracy, this is even more challenging as rank is very sensitive to small perturbations. The numerical accuracy of $\text{float}32$, $1.19e - 7$ [38], cannot be trivially neglected in computing matrix ranks. Thus, in this section we establish some numerical tools for validating our subsequent arguments, and provide rigorous theoretical support for them.

## 3.1 Numerical Rank: Stable Alternative to Rank

The rank of large matrices is known to be unstable: it varies significantly under even small noise perturbations [41]. Matrices perturbed by even small Gaussian noises are almost surely of full rank, regardless of the true rank of the original matrix. Thus in practice we have to use an alternative: we count the number of singular values larger than some given threshold $\epsilon$ as the numerical rank of the matrix. Let $\boldsymbol{W} \in \mathbb{R}^{n \times d}$ be a given matrix. Its numerical rank with tolerance $\epsilon$ is

$$\text{Rank}_\epsilon(\boldsymbol{W}) = \#\{i \in \mathbb{N}_+ : i \leq min\{n, d\}, \sigma_i \geq \epsilon \|\boldsymbol{W}\|_2\}, \tag{4}$$

where $\|\boldsymbol{W}\|_2$ is the $\ell_2$ norm (spectral norm) of matrix $W$, $\sigma_i, i = 1, ..., \min\{n, d\}$ are its singular values, and $\#$ is the counting measurement for finite sets. We can prove that the numerical rank is stable under small perturbations. Based on Weyl inequalities [48], we have the following theorem.

**Theorem 1.** *For any given matrix $\boldsymbol{W}$, almost every tolerance $\epsilon > 0$, and any perturbation matrix $\boldsymbol{D}$, there exists a positive constant $\delta_{\max}(\epsilon)$ such that $\forall \delta \in [0, \delta_{\max}(\epsilon))$, $\text{Rank}_\epsilon(\boldsymbol{W} + \delta \boldsymbol{D}) = \text{Rank}_\epsilon(\boldsymbol{W})$. If $\boldsymbol{W}$ is a low-rank matrix without random perturbations, then there is a $\epsilon_{\max}$ such that for any $\epsilon < \epsilon_{\max}$, $\text{Rank}_\epsilon(\boldsymbol{W} + \delta \boldsymbol{D}) = \text{Rank}_\epsilon(\boldsymbol{W}) = \text{Rank}(\boldsymbol{W})$ for all $\delta \in [0, \delta_{\max}(\epsilon))$.*

This property of the numerical rank metric makes it a suitable tool for investigating the rank behavior of neural networks. Possible small noises can be filtered out in Jacobian matrices of networks by using numerical rank. It is worth mentioning that random matrices no longer have full rank almost surely under the numerical rank. Instead their rank distribution can be inferred from the well-known Marcenko–Pastur distribution [33] of random matrices. So under numerical rank, low-rank matrices will be commonly seen. In this paper, we always use the numerical rank when measuring ranks.

## 3.2 Partial Rank of the Jacobian: Estimating Lower Bound of Lost Rank in Deep Networks

To enable the validation of trend of the network ranks, we propose to compute only the rank of sub-matrices of the Jacobian as an alternative. Those sub-matrices are also the Jacobian matrices with respect to a fixed small patch of inputs. Rigorously, given a function $\boldsymbol{f}$ and its Jacobian $\boldsymbol{J_f}$, we denote partial rank of the Jacobian as the rank of a sub-matrix of the Jacobian that consists of the $j_1$-th, $j_2$-th,...,$j_K$-th column of the original Jacobian

$$\text{PartialRank}(\boldsymbol{J_f}) = \text{Rank}(\text{Sub}(\boldsymbol{J_f}, j_1, ..., j_K)) = \text{Rank}((\partial \boldsymbol{f}_i / \partial \boldsymbol{x}_{j_k})_{d \times K}), \tag{5}$$

where $1 \leq j_1 < \ldots < j_K \leq n$. The partial rank can be efficiently computed and the variance of partial ranks of adjacent sub-networks gives a lower bound on the variance of their ranks. We demonstrate this in Appendix F.

## 3.3 Classification Dimension: Estimating Final Feature Dimension

Measuring the intrinsic dimension of feature manifolds is known to be intractable. So we turn to an approximation procedure. For most classification networks, a linear regression over the final feature manifold decides the final network prediction and accuracy. So we can estimate the intrinsic dimension as the minimum number of principal components in the final feature space to preserve a high classification accuracy. Let $\text{cls} : \mathbb{R}^d \to \mathbb{R}^c$ be the classification predictions based on the final

| Networks | ResNet-18 | ResNet-50 | GluMixer-24 | ResMLP-S24 | Swin-T | ViT-T |
|---|---|---|---|---|---|---|
| ClsDim | 149 | 131 | 199 | 196 | 344 | 109 |
| Ambient Dim. | 512 | 2048 | 384 | 384 | 768 | 192 |

Table 1: Classification dimensions (with respect to 95% classification performance of the ambient feature space $\mathbb{R}^d$) and ambient dimensions of the final feature manifolds of different networks. All networks have low intrinsic dimensions for final features.

feature representation $\boldsymbol{F}(\boldsymbol{x})$, $\mathrm{Pro}_k$ be the operator that project a vector to the subspace spanned by the top-$k$ PCA components of the final feature representations, and $\mathbf{1}_{\mathrm{cond}}$ the indicator for condition cond. The classification dimension is then defined as

$$\mathrm{ClsDim}(\boldsymbol{F}(\mathcal{X})) = \min_k\{k : \mathbb{E}_{(\boldsymbol{x},\boldsymbol{y})\sim\mathbb{P}_{\mathcal{X},\mathcal{Y}}}[\mathbf{1}_{\mathrm{Cls}(\mathrm{Pro}_k(\boldsymbol{F}(\boldsymbol{x})))==\boldsymbol{y}}] \geq 1 - \epsilon\}, \tag{6}$$

which is the minimum dimensionality needed to reconstruct the classification accuracy of the whole model.

## 4 Simple Principle of Rank Diminishing for General Networks

The specific designs of neural networks are vast and diverse, but most of them share two fundamental ingredients of deep networks, *i.e.*, *the chain rule and matrix multiplication*. This provides us a chance to analyze the behavior of deep networks that is architecture-agnostic. So in the theoretical aspect, we will focus on how these two intrinsic structures endow impetus of rank diminishing to neural networks.

A most straightforward observation comes from the basic rule of matrix multiplication that for any two matrices $\boldsymbol{A}$ and $\boldsymbol{B}$, we have $\mathrm{Rank}(\boldsymbol{AB}) \leq \min\{\mathrm{Rank}(\boldsymbol{A}), \mathrm{Rank}(\boldsymbol{B})\}$ [22]. Taking this into the chain rule of differential of $\boldsymbol{J_F} = \boldsymbol{J}_{\boldsymbol{f}^L}\boldsymbol{J}_{\boldsymbol{f}^{L-1}}...\boldsymbol{J}_{\boldsymbol{f}^1}$, we then have $\mathrm{Rank}(\boldsymbol{J}_{\boldsymbol{F}_k}) = \mathrm{Rank}(\boldsymbol{J}_{\boldsymbol{f}^k\circ\boldsymbol{F}_{k-1}}) = \mathrm{Rank}(\boldsymbol{J}_{\boldsymbol{f}^k}\boldsymbol{J}_{\boldsymbol{F}_{k-1}}) \leq \mathrm{Rank}(\boldsymbol{J}_{\boldsymbol{F}_{k-1}}), k = 2,...,L$, which is Eq. (7). We can then get the following principle of rank diminishing.

**Theorem 2** (Principle of Rank Diminishing). *Suppose that each layer $\boldsymbol{f}_i, i = 1,...,L$ of network $\boldsymbol{F}$ is almost everywhere smooth[4] and data domain $\mathcal{X}$ is a manifold, then both the rank of sub-networks and intrinsic dimension of feature manifolds decrease monotonically by depth:*

$$\mathrm{Rank}(\boldsymbol{f}_1) \geq \mathrm{Rank}(\boldsymbol{f}_2 \circ \boldsymbol{f}_1) \geq ... \geq \mathrm{Rank}(\boldsymbol{f}_{L-1} \circ ... \circ \boldsymbol{f}_1) \geq \mathrm{Rank}(\boldsymbol{F}_L), \tag{7}$$

$$\dim(\mathcal{X}) \geq \dim(\mathcal{F}_1) \geq \dim(\mathcal{F}_2) \geq ... \geq \dim(\mathcal{F}_L). \tag{8}$$

This principle describes the behavior of generic neural networks with almost everywhere smooth components, which exhibits the monotonic decreasing (but not strictly) of network ranks and intrinsic dimensionality of feature manifolds.

A flaw of the above principle is that it does not tell whether the rank must decrease. So we need further analysis for the chance of strictly decreasing. A most direct reason to support the strictly decreasing comes from the structural impetus of rank deficiency of numerous network components. Frequently used operations like pooling, downsampling, and dense layer can loose ranks considerably as they explicitly decrease the ambient dimensions of feature representations, or have low rank weight matrices [34]. Specifically, we can have a global criterion for whether a network component will lose ranks as follows.

**Theorem 3** (Structural Impetus of Strictly Decreasing). [5] *Roughly speaking, if almost everywhere on the input feature manifold, there is a direction such that moving along this direction keeps the output invariant, then the intrinsic dimension of the output feature manifold will be strictly lower than that of the input. The maximum number of independent such directions gives a lower bound on the number of lost intrinsic dimensions.*

For example, the ReLU activation gives the same output for all inputs that only differs from each other in the negative parts. Thus the feature manifold after a ReLu activation will lose the dimension

---

[4]It means having arbitrary order gradients except for a zero measure set in the input domain

[5]The rigorous version is given in the Appendix.

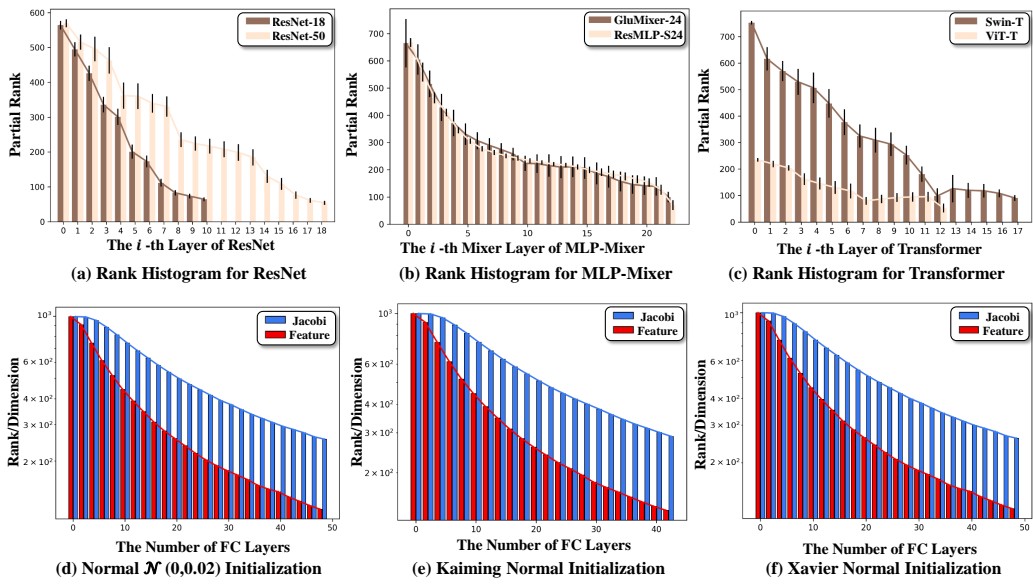

Figure 1: Partial rank of Jacobian matrices of CNN, MLP, and Transformer architecture networks for different layers on ImageNet (top row); rank of Jacobian matrices and feature dimensions of linear MLP network following conditions of Theorem 5 (bottom row). All the models show a similar trend of fast decreasing of ranks as predicted by Theorems 4 and 5.

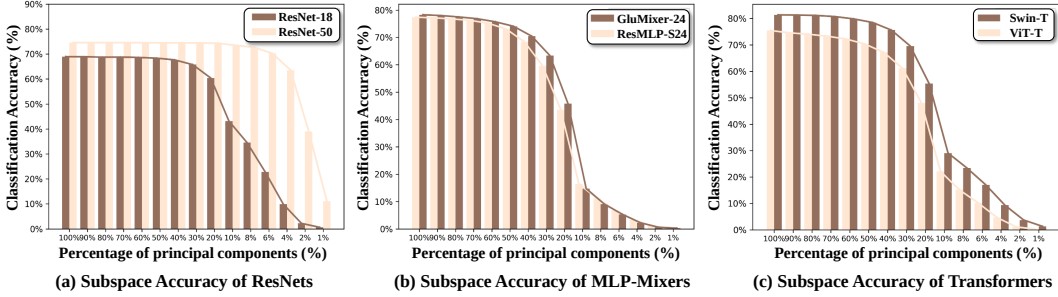

Figure 2: Classification Accuracy (top-1) of using subspaces spanned by top-$k\%$ eigenvectors (principal components) of the final feature manifolds. For all networks a small percentage (see Tab. 1) of eigenvectors is enough to reproduce the classification accuracy of the whole network, indicating a low intrinsic dimension of final feature manifolds. **Note that the $x$-axes are non-linear.**

for those negative parts. For general linear layers (*e.g.,* pure convolution and dense layers), the feature dimensions that belong to the orthogonal complementary spaces of the weights will be lost after applying the linear transformations, as the weight matrices are inactive to changes in their orthogonal complementary spaces.

## 5  Limiting Behavior of Network Ranks for Infinitely Deep Networks

From a theoretical perspective, it is of great interest to consider the limiting behavior—the rank of infinitely deep neural networks. However, it is infeasible to directly consider this problem and give any rigorous analysis for general cases— it is impossible to exhaust the tedious discussion of how specific structures can influence network ranks. Thus we need to concentrate on a simplified but still representative math model of this problem.

**Theory Setup and Its Necessity**    In brief, we will assume in this section that the Jacobi matrices are random matrices independently following some distributions in a fixed Euclidean space. We then consider the limiting behavior of the singular values of series $\{J_{F_n}\}_{n=1}^{\infty}$. Modeling the Jacobi

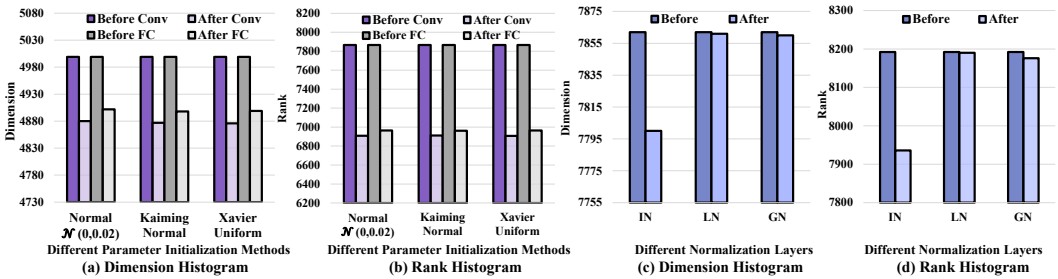

Figure 3: PCA dimension of feature spaces and rank of Jacobian matrix for commonly seen network components under standard Gaussian inputs and randomized weights. Convolution and FC layers tend to lose rank considerably; normalization layers, like InstanceNorm (IN) [45], LayerNorm (LN) [2], and GroupNorm (GN) [51], lose rank modestly. But none can preserve rank.

matrices with random matrices allows us to neglect the influence of specific architectures and reveal the intrinsic property using large number law of probability. While limiting them in the same space is a common practice in analyzing infinitely deep networks [42, 36, 39, 3, 32], as it offers the feasibility to define limits of variables. For example, if the sizes of Jacobi matrices change from time to time, some singular values may suddenly disappear or appear in the series $\{J_{F_n}\}_{n=1}^{\infty}$, thus it is unable to define limits for them.

After this simplification, the problem is still representative for understanding what happens in reality. The transformer, recurrent neural networks, and some very deep CNN architectures with constant width [52] can be viewed as examples of this setting. Another interesting setting is to assume that the Jacobi matrices follow the most simple distribution—Gaussian distributions. This can lead to a much stronger yet intuitive result on network ranks. So we are most interested in the limiting behavior when the Jacobi matrices follow weak regularized distributions and the Gaussian distributions. These two cases then lead to Theorems 4 and 5 correspondingly.

**Theorem 4.** *Let the network be* $F = f^L \circ \cdots \circ f^1$, *and all the ambient dimensions of feature manifolds be the same as the ambient dimension of inputs, i.e.,* $f^k : \mathbb{R}^n \to \mathbb{R}^n, k = 1, \ldots, L$. *Suppose the Jacobian matrix of each layer* $f_i$ *independently follows some distribution* $\mu$, *and* $\mathbb{E}_\mu[\max\{\log \|J_{f^k}^{\pm 1}\|_2, 0\}] < \infty$. *Let* $\sigma_k$ *denote the* $k$-*th largest singular value of* $J_F$. *Then there is an integer* $r < n$ *and positive constants* $\mu_r, \ldots, \mu_n$ *that only depend on* $\mu$ *such that we have for* $\mu$-*almost everywhere,*

$$\frac{\sigma_k}{\|J_F\|_2} \sim \exp(-L\mu_k) \to 0, k = r, \ldots, n, \text{ as } L \to \infty, \tag{9}$$

*meaning that for any tolerance* $\epsilon > 0$, $\mathrm{Rank}_\epsilon(F)$ *drops below* $r + 1$ *as* $L \to \infty$.

**Theorem 5.** *Let the network be* $F = f^L \circ \ldots \circ f^1$, *and all the ambient dimensions of feature manifolds be the same as the ambient dimension of inputs, i.e.,* $f^k : \mathbb{R}^n \to \mathbb{R}^n, k = 1, \ldots, L$. *Suppose that* $J_{f^i}$ *independently follows the standard Gaussian distribution. Let* $\sigma_k$ *denote the* $k$-*th largest singular value of* $J_F$. *Then almost surely*

$$\lim_{L \to \infty} \left(\frac{\sigma_k}{\|J_F\|_2}\right)^{\frac{1}{L}} = \exp \frac{1}{2} \left(\psi(\frac{n-k+1}{2}) - \psi(\frac{n}{2})\right) < 1, k = 2, \ldots, n, \tag{10}$$

*where* $\psi = \Gamma/\Gamma'$ *and* $\Gamma$ *is the Gamma function. That means for a large* $L$ *and any tolerance* $\epsilon$, $\mathrm{Rank}_\epsilon(F)$ *drops to 1 exponentially with speed* $nC^L$, *where* $C < 1$ *is a positive constant that only depends on* $n$.

The proofs of these two theorems rely on the advances of random matrix theory, especially the study of Lyapunov exponents of linear co-cycle systems [43, 50, 18, 40, 29] (see Appendix A for detail). Both of the two theorems reveal that infinitely deep networks have an intrinsic intention to drop ranks rapidly and centralize energy in the largest singular vectors. This intention is purely aroused by the two essential ingredients, chain rule and matrix multiplication, as here we omit the consideration of specific network designs, but focus on the large number law of coupling of massive smooth functions.

These two theorems point out that the ratio of non-largest singular value divided by the largest one will converge to zero with an exponential speed. As a result, the numerical rank of the network will decrease quickly after the layer is deep enough. While this is not the hard rank we use in Theorem 2, we can still use it to infer the low rank structure of feature representations. Let $\boldsymbol{q}_1, \cdots, \boldsymbol{q}_n$ be the singular vectors of $\boldsymbol{J_F}$ corresponding to the singular values in Theorem 4, and $\alpha_i = \langle \Delta \boldsymbol{x} / \| \Delta \boldsymbol{x} \|_2, \boldsymbol{q}_i \rangle$, then when the depth $L \to \infty$ and $\Delta \boldsymbol{x}$ is small, we have $\boldsymbol{F}(\boldsymbol{x} + \Delta \boldsymbol{x}) = \boldsymbol{F}(\boldsymbol{x}) + \boldsymbol{J_F} \Delta \boldsymbol{x} + o(\| \Delta \boldsymbol{x} \|_2) = \sigma_1 \sum_{i=1}^r \alpha_i \boldsymbol{q}_i + \sigma_1 \sum_{j=2}^n \frac{\sigma_j}{\sigma_1} \alpha_j \boldsymbol{q}_j + o(\| \Delta \boldsymbol{x} \|_2) \approx \sigma_1 \sum_{i=1}^r \alpha_i \boldsymbol{q}_i$, which means the neighborhood of data $\boldsymbol{x}$ is mapped into an $r$-dimensional subspace spanned by the singular vectors of the largest singular values.

This can also explain why layer-wise regularization techniques are helpful for training deep networks, as those regularization techniques re-normalize the singular value distributions of deep networks, easing the trend of the ratio $\frac{\sigma_i}{\sigma_1}$ becoming infinitesimal. Specifically, if the data distribution near $\boldsymbol{x}$ is assumed to be standard Gaussian, then the feature distribution is $\mathcal{N}(\boldsymbol{F}(\boldsymbol{x}), \boldsymbol{J_F^T} \boldsymbol{J_F})$, which has a very low rank covariance matrix due to $\frac{\sigma_i}{\sigma_1}, i \geq r$ is tiny. Then applying Batch Normalization will try to pull this feature distribution back to Gaussian with identity covariance, thus ease the trend of feature dimension collapse. This has also been discussed in related work of the Batch Normalization [13, 5]. We will discuss some other techniques to remiss the rank diminishing in the Appendix.

These two theorems are also connected with the famous gradient explosion issue of deep neural networks [4, 37], where the largest singular value of the Jacobian matrix tends to infinity when the layer gets deeper. This issue could be viewed as a special case of Theorem 5 that investigates the behavior of all singular values of deep neural networks. The behavior of network ranks in fact manipulates the well-known gradient explosion issue. Rigorously, we have the following conclusion.

**Corollary 1.** *Under the condition of Theorem 5, then almost surely gradient explosion happens at an exponential speed, i.e., $\log \| \boldsymbol{J_F} \|_2 = \log \sigma_1 \sim \frac{L}{2}(\log 2 + \psi(n/2)) \to \infty$ when $L$ is large.*

## 6 Validating the Rank Behavior

### 6.1 Validating Rank Diminishing by Depth

In this section, we numerically validate our theory in three types of architectures of benchmark deep neural networks, CNNs, MLPs, and Transformers, in the ImageNet [14] data domain. Information of those networks is listed in (Appendix) Tab. A2. For validating the tendency of network rank of Jacobian matrices, we use the numerical rank of sub-matrices of Jacobian on the central $16 \times 16 \times 3$ image patch of input images. We report the results of other choices of patches in the Appendix. Details of the experiment setup can be found in Appendix E.

**Rank Diminishing of Jacobians** As is discussed in Sec. 3.2 and (Appendix) Lemma 2, the partial rank of the Jacobian is a powerful weapon for us to detect the behavior of huge Jacobian matrices, which are infeasible to compute in practice. The decent value of partial ranks of adjacent sub-networks provides a lower bound to that of full ranks of them. Fig. 1 (a,b,c) report the partial rank of Jacobian matrices of three types of architectures, where we can find consistent diminishing of partial ranks in each layer, indicating a larger rank losing for the full rank of Jacobian matrices.

**Implicit Impetus** Theorem 5 gives an exponential speed of rank decent by layers. We find that it corresponds well with practice. We investigate this exponential law in a toy network of MLP-50, which is composed of 50 dense layers, each with 1,000 hidden units. The MLP-50 network takes Gaussian noise vectors of $\mathbb{R}^{1000}$ as inputs, and returns a prediction of 1,000 categories. As all the feature manifolds are linear subspaces in this case, their intrinsic dimensions can be directly measured by the numerical rank of their covariance matrices. We report the full rank of Jacobian matrices and intrinsic dimensions of feature manifolds under three different randomly chosen weights in Fig. 1 (d,e,f). Due to the digital accuracy of float32, we stop calculation in each setting when the absolute values of elements of the matrices are lower than $1.19e-7$. We can find standard curves of exponential laws in all cases for both ranks of Jacobian and intrinsic dimensions of features.

**Structural Impetus** We validate the structural impetus in Fig. 3. To give an estimation for general cases, here we use Gaussian noises with the size of $128 \times 8 \times 8$ as inputs, and randomize weights of the network components to be validated. We plug those components into a simple fully-connected (FC)

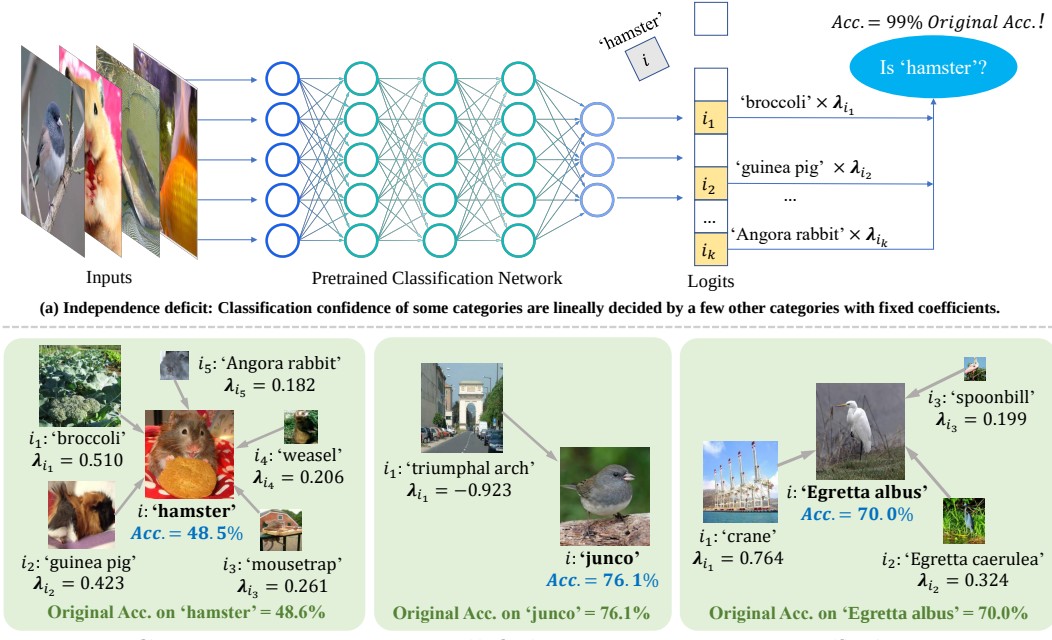

**(a) Independence deficit: Classification confidence of some categories are lineally decided by a few other categories with fixed coefficients.**

| (b) ResNet-50 | (c) GluMixer-24 | (d) Swin-T |
|---|---|---|

Figure 4: Independence deficit. Classification confidence of some ImageNet categories are lineally decided by a few other categories with fixed coefficients in the whole data domain. We illustrate this phenomenon in (a). Here we present some results from ResNet-50, GluMixer-24, and Swin-T. In the (b,c,d) we illustrate the categories of $i_1, ..., i_k$ (in the surrounding) to linearly decide category $i$ (in the center) and their corresponding weights $\lambda_{i_1}, ..., \lambda_{i_k}$. The classification accuracy on the validation set of using Eq. (12), instead of the true logits, to predict the label is reported in **blue** (if tested on positive samples only, the accuracy rates are 98%, 90%, 82% for cases in (b,c,d) correspondingly). For comparison, the original accuracy for the corresponding categories are reported in **green**. We can find that 1) a few other categories can decide the confidence of the target category $i$; 2) some very irrelevant categories contribute the largest weights. For example in (c), the logits of class 'junco' is the negative of 'triumphal arch'. Both of them indicate a rather drastic competition of different categories for independent representations in final features due to the tight rank budgets.

layer of 8,192 hidden units. As the structure is simple, we directly measure the intrinsic dimension of feature spaces and the full rank of Jacobian matrices before and after the features pass the network components to be measured. The dimension is determined by the number of PCA eigenvalues [23, 49] larger than $1.19e - 7 \times N \times \sigma_{\max}$, where $N$ is the number of PCA eigenvalues, and $\sigma_{\max}$ is the largest PCA eigenvalue. The batch size is set to 5,000. We find that the convolution (the kernel size is $3 \times 3$) and FC layers (the weight size is 8,192) tend to lose rank considerably, while different normalization layers also lose rank modestly. But none of them can preserve rank invariant.

## 6.2 Validating Low-Rank Terminal Spaces

This section seeks evidence to support a low-rank terminal feature space of deep networks, which can then support the significant diminishing of ranks in the previous layers. The evidence is consisted of a numerical validation that investigates the classification dimension of the terminal feature layer, together with a semantic validation that reveals the independence deficit of deep networks.

**Numerical Validation**    To get an estimation of how many dimensions remain in the final feature representation, we measure the classification dimension in Fig. 2 and Tab. 1. We report the classification accuracy produced by projecting final feature representations to its top $k\%$ eigenvectors in Fig. 2. We choose a threshold of $\epsilon$ such that this procedure can reproduce 95% of the original accuracy of the network. The corresponding ClsDim is reported in Tab. 1. As discussed in Sec. 3.3, this gives an estimation of the intrinsic dimension of the final feature manifold. We can find a universal low-rank structure for all types of networks.

**Semantic Validation**   We want to show that there are only a few independent representations to decide the classification scores for all the 1,000 categories of ImageNet. Specifically, can we predict the outputs of the network for some categories based on the outputs for a few other categories, as illustrated in Fig. 4 (a)? And if we can, will those categories be strongly connected to each other? A surprising fact is that, we can find many counter examples of irrelevant categories dominating the network outputs for given categories regarding various network architectures. This interesting phenomenon indicates a rather drastic competing in the final feature layer for the tight rank budgets of all categories, which yields non-realistic dependencies of different categories.

To find the dependencies of categories in final features, we can solve the following Lasso problem [44],

$$\boldsymbol{\lambda}^* = \arg\min_{\boldsymbol{\lambda}_i = -1} \mathbb{E}_{\boldsymbol{x}}[\|\boldsymbol{\lambda}^T \boldsymbol{W} \boldsymbol{F}(\boldsymbol{x})\|_2^2] + \eta\|\boldsymbol{\lambda}\|_1, \tag{11}$$

where $\boldsymbol{F}(\boldsymbol{x}) \in \mathbb{R}^{1000}$ is the slice of network from inputs to the final feature representation, $\boldsymbol{x}$ is the sample from ImageNet $\mathcal{X}$, and $\boldsymbol{W}$ is the final dense layer. The solution $\boldsymbol{\lambda}^*$ will be a sparse vector, with $k$ non-zero elements $\lambda_{i_1} \geq \lambda_{i_2} \geq ... \geq \lambda_{i_k}, k \ll 1000$. We can then get

$$\text{logits}(\boldsymbol{x}, i) \approx \boldsymbol{\lambda}_{i_1}\text{logits}(\boldsymbol{x}, i_1) + ... + \boldsymbol{\lambda}_{i_k}\text{logits}(\boldsymbol{x}, i_k), i \notin \{i_1, ..., i_k\}, k \ll 1000, \forall \boldsymbol{x} \in \mathcal{X}, \tag{12}$$

where $\text{logits}(\boldsymbol{x}, i_j), j = 1, ..., k$ is the logits of network for category $i_j$, *i.e.*, $\text{logits}(\boldsymbol{x}, i_j) = \boldsymbol{W}_{i_j}\boldsymbol{F}(\boldsymbol{x})$. It is easy to see that outputs for category $i$ are linearly decided by outputs for $i_1, ..., i_k$ and are dominated by outputs for $i_1$.

Fig. 4 demonstrates the solutions of Eq. (12) for three different categories in ImageNet with $\eta = 20$, and network architectures ResNet-50, GluMixer-24, and Swin-T. The results are surprising. It shows that many categories of the network predictions are in fact 'redundant', as they are purely decided by the predictions of the other categories with simple linear coefficients. In this case, the entanglement of different categories cannot be avoided, thus the network may perform poorly under domain shift. An even more surprising finding is that, some very irrelevant categories hold the largest weights when deciding the predictions of the redundant categories, which means that the networks just neglect the unique representations of those categories in training and yield over-fitting when predicting them.

## 7   Related Work

Previous studies of rank deficiency in deep neural networks follow two parallel clues. One is the study of rank behavior in specific neural network architectures. Dong et al. [15] studies deep networks consisting of pure self-attention networks, and proves that they converge exponentially to a rank-1 matrix under the assumption of globally bounded weight matrices. Daneshmand et al. [13] studies the effect of BatchNorm on MLPs and shows that BatchNorm can prevent drastic diminishing of network ranks in some small networks and datasets. Both of those works avoid directly validating the behavior of network ranks in intermediate layers due to the lacking of efficient numerical tools. An independent clue is the study of implicit self-regularization, which finds that weight matrices tend to lose ranks after training. Martin and Mahoney [34] studies this phenomenon in infinitely-wide, over-parametric neural networks with tools from random matrix theory. Arora et al. [1] studies this phenomenon in deep matrix decomposition. Those works focus on the theoretical behavior of rank of weight matrices induced by the training instead of network ranks.

## 8   Conclusion

This paper studies the rank behavior of deep neural networks. In contrast to previous work, we focus on directly validating rank behavior with deep neural networks of diverse benchmarks and various settings for real scenarios. We first formalize the analysis and measurement of network ranks. Then under the proposed numerical tools and theoretical analysis, we demonstrate the universal rank diminishing of deep neural networks from both empirical and theoretical perspectives. We further support the rank-deficient structure of networks by revealing the independence deficit phenomenon, where network predictions for a category can be linearly decided by a few other, even irrelevant categories. The results of this work may advance understanding of the behavior of fundamental network architectures and provide intuition for a wide range of work pertaining to network ranks.

## Acknowledgement

The authors thank Prof. Dangzheng Liu of the University of Science and Technology of China for disucssion. This work was supported by National Key R&D Program of China under Grant 2020AAA0105702, National Natural Science Foundation of China (NSFC) under Grants 62225207 and U19B2038, and the University Synergy Innovation Program of Anhui Province under Grants GXXT-2019-025.

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
