# Appendix

## Contents

## A  Proofs

### A.1  Proof to Lemma 1

This Lemma is the direct result of the *rank theorem of manifolds*.

**Theorem 6** (Rank Theorem [25]). *Suppose $\boldsymbol{f} : \mathcal{M} \to \mathcal{N}$ is a smooth function from $m$-dimensional manifold $\mathcal{M}$ to $n$-dimensional manifold $\mathcal{N}$, and $\mathrm{Rank}_{\mathcal{M},\mathcal{N}}(\boldsymbol{J_f}) = r$. Then for each $\boldsymbol{x} \in \mathcal{M}$, there exists a smooth chart $(\mathcal{U}, \boldsymbol{m})$ around $\boldsymbol{x}$ and a smooth chart $(\mathcal{V}, \boldsymbol{n})$ around $\boldsymbol{f}(\boldsymbol{x})$, such that*

$$\boldsymbol{n} \circ \boldsymbol{f} \circ \boldsymbol{m}^{-1} : \boldsymbol{m}(\mathcal{U}) \subset \mathbb{R}^m \to \boldsymbol{n}(\mathcal{V}) \subset \mathbb{R}^n \tag{A13}$$

*is given by $\boldsymbol{n} \circ \boldsymbol{f} \circ \boldsymbol{m}^{-1}(\boldsymbol{x}_1, \boldsymbol{x}_2, \cdots, \boldsymbol{x}_m) = (\boldsymbol{x}_1, \boldsymbol{x}_2, \cdots, \boldsymbol{x}_m, 0, \cdots, 0)$.*

The rank of a function $\mathrm{Rank}_{\mathcal{M},\mathcal{N}}$ defined on the manifold is the rank under local chart systems of the input manifold $\mathcal{M}$ and output manifold $\mathcal{N}$. Let $\phi$ be the chart for point $\boldsymbol{x} \in \mathcal{O} \subset \mathcal{M}$, and the identity map be the chart for point $\boldsymbol{f}(\boldsymbol{x}) \in \mathbb{R}^d$. Then it is easy to find that

$$\mathrm{Rank}(\boldsymbol{I}^{-1} \circ \boldsymbol{f} \circ \phi) = \mathrm{Rank}(\boldsymbol{f} \circ \phi) = \mathrm{Rank}_{\mathcal{M},\mathcal{N}}(\boldsymbol{f}). \tag{A14}$$

Then by Theorem 6, we know that

$$\dim(\boldsymbol{f}(\mathcal{X})) = \mathrm{Rank}_{\mathcal{M},\mathcal{N}}(\boldsymbol{f}) = \mathrm{Rank}(\boldsymbol{f} \circ \phi) \leq \mathrm{Rank}(\boldsymbol{f}) = r. \tag{A15}$$

The last equal sign comes from the rank inequality of matrix multiplication $\mathrm{Rank}(\boldsymbol{AB}) \leq \min\{\mathrm{Rank}(\boldsymbol{A}), \mathrm{Rank}(\boldsymbol{B})\}$, which we will discuss later.

## A.2   Proof to Theorem 1

Proof to this Theorem needs Weyl's inequalities [57] for singular values of sum of matrices.

**Theorem 7** (Weyl's inequalities). *Let $\boldsymbol{A}, \boldsymbol{B}$ be $p \times n$ complex matrices, $\sigma_i(\cdot)$ be the $i$-th largest singular value of the matrix. Then*

$$|\sigma_i(\boldsymbol{A} + \boldsymbol{B}) - \sigma_i(\boldsymbol{B})| \leq \sigma_1(\boldsymbol{B}), 1 \leq i \leq p, n. \tag{A16}$$

Let $p$ be the number of singular values of $\boldsymbol{W}$ and $\boldsymbol{D}$. By this theorem, we have

$$\sigma_i(\boldsymbol{W}) - \delta\sigma_1(\boldsymbol{D}) \leq \sigma_i(\boldsymbol{W} + \delta\boldsymbol{D}) \leq \sigma_i(\boldsymbol{W}) + \delta\sigma_1(\boldsymbol{D}), i = 1, \cdots, p. \tag{A17}$$

To measure the numerical rank, we need to estimate the relative quantities of singular values, which are,

$$\frac{\sigma_i(\boldsymbol{W}) - \delta\sigma_1(\boldsymbol{D})}{\sigma_1(\boldsymbol{W}) + \delta\sigma_1(\boldsymbol{D})} \leq \frac{\sigma_i(\boldsymbol{W} + \delta\boldsymbol{D})}{\sigma_1(\boldsymbol{W} + \delta\boldsymbol{D})} \leq \frac{\sigma_i(\boldsymbol{W}) + \delta\sigma_1(\boldsymbol{D})}{\sigma_1(\boldsymbol{W}) - \delta\sigma_1(\boldsymbol{D})}, i = 2, \cdots, p. \tag{A18}$$

Now assume that $\epsilon$ does not belong to the following set (which is a zero measure set in $\mathbb{R}_+$)

$$\Sigma_{\boldsymbol{W}} = \left\{ \frac{\sigma_i(\boldsymbol{W})}{\sigma_1(\boldsymbol{W})} : i = 2, \cdots, p \right\}, \tag{A19}$$

and $\mathrm{Rank}_\epsilon(\boldsymbol{W}) = r$. We know that

$$\frac{\sigma_i(\boldsymbol{W})}{\sigma_1(\boldsymbol{W})} > \epsilon, i = 2, \cdots, r; \frac{\sigma_i(\boldsymbol{W})}{\sigma_1(\boldsymbol{W})} < \epsilon, i = r+1, \cdots, p. \tag{A20}$$

Thus, we have that $\forall \delta < \delta_{\max}$,

$$\frac{\sigma_i(\boldsymbol{W} + \delta\boldsymbol{D})}{\sigma_1(\boldsymbol{W} + \delta\boldsymbol{D})} \geq \frac{\sigma_i(\boldsymbol{W}) - \delta\sigma_1(\boldsymbol{D})}{\sigma_1(\boldsymbol{W}) + \delta\sigma_1(\boldsymbol{D})} > \epsilon, i = 2, \cdots, r, \tag{A21}$$

$$\frac{\sigma_i(\boldsymbol{W})}{\sigma_1(\boldsymbol{W})} \leq \frac{\sigma_i(\boldsymbol{W}) + \delta\sigma_1(\boldsymbol{D})}{\sigma_1(\boldsymbol{W}) - \delta\sigma_1(\boldsymbol{D})} < \epsilon, i = r+1, \cdots, p, \tag{A22}$$

provided that

$$\begin{aligned}
\delta_{\max} = \min\{ &\frac{1}{\sigma_1(D)} \left( \frac{\sigma_r(\boldsymbol{W}) + \sigma_1(\boldsymbol{W})}{\epsilon + 1} - \sigma_1(\boldsymbol{W}) \right), \frac{\sigma_r(\boldsymbol{W})}{2\sigma_1(\boldsymbol{D})}, \\
&\frac{1}{\sigma_1(D)} \left( \sigma_1(\boldsymbol{W}) - \frac{\sigma_{r+1}(\boldsymbol{W}) + \sigma_1(\boldsymbol{W})}{\epsilon + 1} \right), \frac{\sigma_1(\boldsymbol{W})}{2\sigma_1(\boldsymbol{D})} \}.
\end{aligned} \tag{A23}$$

Thus we can conclude

$$\mathrm{Rank}_\epsilon(\boldsymbol{W} + \delta\boldsymbol{D}) = \mathrm{Rank}_\epsilon(\boldsymbol{W}), \forall \delta \in [0, \delta_{\max}). \tag{A24}$$

When $\mathrm{Rank}(\boldsymbol{W}) = r < p$, it is then easy to see if $\epsilon < \frac{\sigma_r(\boldsymbol{W})}{\sigma_1(\boldsymbol{W})}$, we have

$$\frac{\sigma_i(\boldsymbol{W})}{\sigma_1(\boldsymbol{W})} > \epsilon, i = 2, \cdots, r; \frac{\sigma_i(\boldsymbol{W})}{\sigma_1(\boldsymbol{W})} = 0 < \epsilon, i = r+1, \cdots, p. \tag{A25}$$

Thus setting $\epsilon_{\max} = \frac{\sigma_r(\boldsymbol{W})}{\sigma_1(\boldsymbol{W})}$, we can always have $\delta_{\max}$ acquired by Eq. (A23), such that

$$\mathrm{Rank}_\epsilon(\boldsymbol{W}) = \mathrm{Rank}(\boldsymbol{W}) = \mathrm{Rank}_\epsilon(\boldsymbol{W} + \delta\boldsymbol{D}), \forall \delta \in [0, \delta_{\max}). \tag{A26}$$

### A.3 Proof to Lemma 2

Let $\boldsymbol{A}_i$ be the $i$-th column of matrix $\boldsymbol{A}$. Given two matrices $\boldsymbol{A} \in \mathbb{R}^{m \times n}, \boldsymbol{B} \in \mathbb{R}^{n \times d}$, we have

$$\boldsymbol{AB} = (\boldsymbol{AB}_1, \cdots \boldsymbol{AB}_d). \tag{A27}$$

Thus for any $1 \leq i_1 < \cdots < i_K \leq d$,

$$(\boldsymbol{AB})_{i_1, \cdots, i_K} = (\boldsymbol{AB}_{i_1}, \cdots, \boldsymbol{AB}_{i_K}) = \boldsymbol{A}(\boldsymbol{B})_{i_1, \cdots, i_K}. \tag{A28}$$

By the rank theorem [26] of matrices, we have

$$\mathrm{Rank}((\boldsymbol{AB})_{i_1, \cdots, i_K}) = \mathrm{Rank}((\boldsymbol{B})_{i_1, \cdots, i_K}) - \dim(\mathrm{Ker}(\boldsymbol{A}) \cap \mathrm{Im}((\boldsymbol{B})_{i_1, \cdots, i_K})), \tag{A29}$$

and

$$\mathrm{Rank}(\boldsymbol{AB}) = \mathrm{Rank}(\boldsymbol{B}) - \dim(\mathrm{Ker}(\boldsymbol{A}) \cap \mathrm{Im}(\boldsymbol{B})). \tag{A30}$$

As $(\boldsymbol{B})_{i_1, \cdots, i_K} \subset \boldsymbol{B}$, it is straightforward to get that

$$\mathrm{Im}((\boldsymbol{B})_{i_1, \cdots, i_K})) \subset \mathrm{Im}(B). \tag{A31}$$

Thus

$$\mathrm{Ker}(\boldsymbol{A}) \cap \mathrm{Im}((\boldsymbol{B})_{i_1, \cdots, i_K}) \subset \mathrm{Ker}(\boldsymbol{A}) \cap \mathrm{Im}(\boldsymbol{B}). \tag{A32}$$

Then we have

$$\begin{aligned} \mathrm{Rank}(\boldsymbol{B}) - \mathrm{Rank}(\boldsymbol{AB}) = \dim(\mathrm{Ker}(\boldsymbol{A}) \cap \mathrm{Im}(\boldsymbol{B}) &\geq \dim(\mathrm{Ker}(\boldsymbol{A}) \cap \mathrm{Im}((\boldsymbol{B})_{i_1, \cdots, i_K})) \\ &= \mathrm{Rank}((\boldsymbol{B})_{i_1, \cdots, i_K})) - \mathrm{Rank}((\boldsymbol{AB})_{i_1, \cdots, i_K}) \geq 0. \end{aligned} \tag{A33}$$

Note that $\mathrm{Rank}(\boldsymbol{f}_2 \circ \boldsymbol{f}_1) = \mathrm{Rank}(\boldsymbol{J}_{\boldsymbol{f}_2} \boldsymbol{J}_{\boldsymbol{f}_1})$. Then we complete the proof.

### A.4 Proof to Theorem 2

The key to this principle is the rank theorem of matrices [26], which is

$$\mathrm{Rank}(\boldsymbol{AB}) = \mathrm{Rank}(\boldsymbol{B}) - \dim(\mathrm{Ker}(\boldsymbol{A}) \cap \mathrm{Im}(\boldsymbol{B})). \tag{A34}$$

Note that $\mathrm{Rank}(\boldsymbol{AB}) = \mathrm{Rank}(\boldsymbol{B}^T \boldsymbol{A}^T)$ and $\mathrm{Rank}(\boldsymbol{A}^T) = \mathrm{Rank}(\boldsymbol{A})$. Then we have

$$\begin{aligned} \mathrm{Rank}(\boldsymbol{AB}) = \mathrm{Rank}(\boldsymbol{B}^T \boldsymbol{A}^T) &= \mathrm{Rank}(\boldsymbol{A}^T) - \dim(\mathrm{Ker}(\boldsymbol{B}^T) \cap \mathrm{Im}(\boldsymbol{A}^T)) \\ &= \mathrm{Rank}(\boldsymbol{A}) - \dim(\mathrm{Ker}(\boldsymbol{B}^T) \cap \mathrm{Im}(\boldsymbol{A}^T)). \end{aligned} \tag{A35}$$

The dimension of a linear subspace will at least be zero, thus the above equations suggest

$$\mathrm{Rank}(\boldsymbol{AB}) \leq \mathrm{Rank}(\boldsymbol{B}), \mathrm{Rank}(\boldsymbol{AB}) \leq \mathrm{Rank}(\boldsymbol{A}). \tag{A36}$$

Applying this argument to the chain rule of differentials then yields the conclusion. Further using Lemma 1 gives the diminishing of intrinsic dimensions of feature manifolds.

### A.5 Proof to Theorem 3

We first give the rigorous version of this theorem as follows.

**Theorem 8.** *Let $\boldsymbol{e}_{\boldsymbol{x}}$ be the exponential map from a small neighborhood $\mathcal{U}_{\boldsymbol{x}}$ of point $\boldsymbol{x}$ on the input feature manifold to its tangent space at $\boldsymbol{x}$, and $\boldsymbol{v}_{\boldsymbol{x}} = \boldsymbol{e}_{\boldsymbol{x}}(\boldsymbol{x})$. Let $\mathcal{X}$ be the input manifold, $s = \dim(\mathcal{X})$, $\boldsymbol{f}^i$ be the layer network, $r = \mathrm{Rank}(\boldsymbol{f}^i)$, and $\boldsymbol{f}^i(\mathcal{X})$ be the output manifold. If for almost everywhere on the input feature manifold, there is a unit vector $\boldsymbol{v} \in \boldsymbol{e}_{\boldsymbol{x}}(\mathcal{U}_{\boldsymbol{x}})$, such that the layer network $\boldsymbol{f}^i$ satisfies*

$$\lim_{t \to 0} \frac{\|\boldsymbol{f}^i \circ \boldsymbol{e}_{\boldsymbol{x}}^{-1}(\boldsymbol{v}_{\boldsymbol{x}}) - \boldsymbol{f}^i \circ \boldsymbol{e}_{\boldsymbol{x}}^{-1}(\boldsymbol{v}_{\boldsymbol{x}} + t\boldsymbol{v})\|_2}{t} = 0, \tag{A37}$$

*then $\dim(\boldsymbol{f}^i(\mathcal{X})) < s$. If the number of such independent $\boldsymbol{v}$ in $\boldsymbol{e}_{\boldsymbol{x}}(\mathcal{U}_{\boldsymbol{x}})$ is $k$, then $\dim(\boldsymbol{f}^i(\mathcal{X})) \leq s - k$.*

Now we prove this theorem.

Note that Eq. (A37) implies
$$\boldsymbol{J}_{\boldsymbol{e}_{\boldsymbol{x}}^{-1}}\boldsymbol{v} \in \mathrm{Ker}(\boldsymbol{J}_{\boldsymbol{f}^i}). \tag{A38}$$

As it is also easy to see
$$\boldsymbol{J}_{\boldsymbol{e}_{\boldsymbol{x}}^{-1}}\boldsymbol{v} \in \mathrm{Im}(\boldsymbol{J}_{\boldsymbol{e}_{\boldsymbol{x}}^{-1}}), \tag{A39}$$

we can conclude
$$\boldsymbol{0} \neq \boldsymbol{J}_{\boldsymbol{e}_{\boldsymbol{x}}^{-1}}\boldsymbol{v} \in \mathrm{Ker}(\boldsymbol{J}_{\boldsymbol{f}^i}) \cap \mathrm{Im}(\boldsymbol{J}_{\boldsymbol{e}_{\boldsymbol{x}}^{-1}}), \tag{A40}$$

where $\boldsymbol{0} \neq \boldsymbol{J}_{\boldsymbol{e}_{\boldsymbol{x}}^{-1}}\boldsymbol{v}$ comes from the full rank property of exponential map and its inverse. Thus we have
$$\dim(\mathrm{Ker}(\boldsymbol{J}_{\boldsymbol{f}^i}) \cap \mathrm{Im}(\boldsymbol{J}_{\boldsymbol{e}_{\boldsymbol{x}}^{-1}})) \geq 1. \tag{A41}$$

Specifically, if linearly independent $\boldsymbol{v}_1, \cdots, \boldsymbol{v}_k$ satisfy Eq. (A37), we can conclude
$$\boldsymbol{0} \neq \boldsymbol{J}_{\boldsymbol{e}_{\boldsymbol{x}}^{-1}}\boldsymbol{v}_i \in \mathrm{Ker}(\boldsymbol{J}_{\boldsymbol{f}^i}) \cap \mathrm{Im}(\boldsymbol{J}_{\boldsymbol{e}_{\boldsymbol{x}}^{-1}}), i = 1, \cdots, k. \tag{A42}$$

As $\boldsymbol{J}_{\boldsymbol{e}_{\boldsymbol{x}}^{-1}}$ has full rank due to the property of exponential map, we know that $\boldsymbol{J}_{\boldsymbol{e}_{\boldsymbol{x}}^{-1}}\boldsymbol{v}_i, i = 1, \cdots, k$ are linearly independent. Then
$$\dim(\mathrm{Ker}(\boldsymbol{J}_{\boldsymbol{f}^i}) \cap \mathrm{Im}(\boldsymbol{J}_{\boldsymbol{e}_{\boldsymbol{x}}^{-1}})) \geq k. \tag{A43}$$

Thus the rank theorem of matrices [26] reads
$$\mathrm{Rank}(\boldsymbol{J}_{\boldsymbol{f}^i \circ \boldsymbol{e}_{\boldsymbol{x}}^{-1}}) = \mathrm{Rank}(\boldsymbol{J}_{\boldsymbol{f}^i}\boldsymbol{J}_{\boldsymbol{e}_{\boldsymbol{x}}^{-1}}) = \mathrm{Rank}(\boldsymbol{J}_{\boldsymbol{e}_{\boldsymbol{x}}^{-1}}) - \dim(\mathrm{Ker}(\boldsymbol{J}_{\boldsymbol{f}^i}) \cap \mathrm{Im}(\boldsymbol{J}_{\boldsymbol{e}_{\boldsymbol{x}}^{-1}})) \tag{A44}$$
$$= s - \dim(\mathrm{Ker}(\boldsymbol{J}_{\boldsymbol{f}^i}) \cap \mathrm{Im}(\boldsymbol{J}_{\boldsymbol{e}_{\boldsymbol{x}}^{-1}})) \leq s - k. \tag{A45}$$

Combining this result with Theorem 6 proves our result.

## A.6 Proof to Theorem 4

The proof to this theorem relies on the existence of Lyapunov exponents of dynamic systems. Given a linearized dynamic system
$$\dot{\boldsymbol{v}}(t) = \boldsymbol{X}_t \boldsymbol{v}, \ \boldsymbol{v}(0) = \boldsymbol{v}_0 \in \mathbb{R}^n, \tag{A46}$$

its (largest) Lyapunov exponent is defined as
$$\lambda = \limsup_{t \to \infty} \frac{1}{t}\|\boldsymbol{v}\|_2. \tag{A47}$$

Further, for a sequence of subspace $\mathcal{L}_h \subset \mathcal{L}_{r-1} \subset \cdots \subset \mathcal{L}_1 \subset \mathcal{L}_0 = \mathbb{R}^n$, we can define the corresponding Lyapunov exponents of all those subspaces as
$$\lambda_i = \lim_{t \to \infty} \frac{1}{t}\log\|\boldsymbol{v}\|_2, \ i = 1, \cdots, h+1, \ \boldsymbol{v}_0 \in \mathcal{L}_{i-1}\backslash\mathcal{L}_i, \tag{A48}$$

and we have
$$\lambda = \lambda_1 > \lambda_2 > \cdots > \lambda_h. \tag{A49}$$

It may be surprising to find that such Lyapunov exponents exist, as $\boldsymbol{v}_0$ can traverse the entire subspace $\mathcal{L}_{i-1}\backslash\mathcal{L}_i$. We will demonstrate the existence of the Lyapunov exponents for our case later in Sec. A.6.3, which is the classical results from the Furstenberg-Kesten theorem [19] and multiplicative ergodic theorem [47]. Before that, we will first assume the existence of those Lyapunov exponents for simplicity of analysis.

Now consider the case of function couplings
$$\boldsymbol{F} = \boldsymbol{f}^L \circ \cdots \circ \boldsymbol{f}^2 \circ \boldsymbol{f}^1, \tag{A50}$$

which has the Jacobian matrix
$$\boldsymbol{J}_{\boldsymbol{F}} = \boldsymbol{J}_{\boldsymbol{f}^L}\boldsymbol{J}_{\boldsymbol{f}^{L-1}} \cdots \boldsymbol{J}_{\boldsymbol{f}^2}\boldsymbol{J}_{\boldsymbol{f}^1}. \tag{A51}$$

Apparently, the following dynamic system induces the Jacobi matrix of $\boldsymbol{F}$,
$$\dot{\boldsymbol{v}}(t) = \boldsymbol{J}_{\boldsymbol{f}^t}\boldsymbol{v}, \ \boldsymbol{v}(0) = \boldsymbol{v}_0 \in \mathbb{R}^n, \ t = 1, \cdots, L. \tag{A52}$$

Thus its Lyapunov exponents are given by

$$\lambda_i = \lim_{L \to \infty} \frac{1}{L}\log\|\boldsymbol{J}_{\boldsymbol{F}}\boldsymbol{v}_0\|_2, \ i = 1, \cdots, h+1, \ \boldsymbol{v}_0 \in \mathcal{L}_{i-1}\backslash\mathcal{L}_i, \tag{A53}$$

for a chain of subspaces $\{0\} = \mathcal{L}_{h+1} \subset \mathcal{L}_h \subset \mathcal{L}_{h-1} \subset \cdots \subset \mathcal{L}_1 \subset \mathcal{L}_0 = \mathbb{R}^n$.

### A.6.1 Lyapunov exponents are limits of logarithms of subspace spectral norm divided by layer depth $L$

We first demonstrate that the Lyapunov exponents are limits of logarithm of the spectral norm of $F$ on $\mathcal{L}_{i-1}\backslash\mathcal{L}_i$ divided by layer depth $L$ when $L \to \infty$, for $i = 1, ..., r$.

It is easy to see

$$\frac{1}{L}\log\|v_0\|_2\|J_F\frac{v_0}{\|v_0\|_2}\|_2 = \frac{1}{L}(\log\|v_0\|_2 + \log\|J_F\frac{v_0}{\|v_0\|_2}\|_2). \tag{A54}$$

When $L \to \infty$, $\frac{1}{L}\log\|v_0\|_2 \to 0$ for any $v_0$, we have

$$\lambda_i = \lim_{L\to\infty}\frac{1}{L}\log\|J_F v_0\|_2, \ \|v_0\|_2 = 1, \ v_0 \in \mathcal{L}_{i-1}\backslash\mathcal{L}_i. \tag{A55}$$

Let

$$\lambda_i^L = \sup_{\|v_0\|_2=1, v_0\in\mathcal{L}_{i-1}\backslash\mathcal{L}_i}\frac{1}{L}\log\|J_F v_0\|_2. \tag{A56}$$

Note that

$$\sup_{\|v_0\|_2=1, v_0\in\mathcal{L}_{i-1}\backslash\mathcal{L}_i}\frac{1}{L}\log\|J_F v_0\|_2 = \frac{1}{L}\log\sup_{\|v_0\|_2=1, v_0\in\mathcal{L}_{i-1}\backslash\mathcal{L}_i}\|J_F v_0\|_2, \tag{A57}$$

and

$$\sup_{\|v_0\|_2=1, v_0\in\mathcal{L}_{i-1}\backslash\mathcal{L}_i}\|J_F v_0\|_2 = \|J_F\|_{2,i}, \tag{A58}$$

where $\|\cdot\|_{2,i}$ denote the spectral norm of a linear operator constrained on $\mathcal{L}_{i-1}\backslash\mathcal{L}_i$. Then we have

$$\lambda_i^L = \frac{1}{L}\log\|J_F\|_{2,i}. \tag{A59}$$

Let $e_{i_1}, \cdots, e_{i_k}$ be a set of standard orthogonal basis of $i_k$ dimensional subspace $\mathcal{L}_{i-1}\backslash\mathcal{L}_i$. If the Lyapunov exponents exist, by Eq. (A55) we have for any $\epsilon > 0$, there is $N \in \mathbb{N}$ such that for all $L > N$,

$$\lambda_i - \frac{\epsilon}{2} \leq \frac{1}{L}\log\|J_F e_j\|_2 \leq \lambda_i + \frac{\epsilon}{2}, j = 1, \cdots, i_k. \tag{A60}$$

Let $v = \sum_{j=1}^{i_k}\alpha_j e_j \in \mathcal{L}_{i-1}\backslash\mathcal{L}_i$, where $\alpha_j, i = 1, \cdots, i_k, \sum_{j=1}^{i_k}\alpha_j^2 = 1$ is the coordinate of unit vector $v$ under the basis $e_j, j = 1, \cdots, i_k$. Assume that $\|J_F e_1\|_2 \geq \|J_F e_j\|_2, j = 2, \cdots, i_k$. We

then have $\frac{|\alpha_j|\|\boldsymbol{J_F}\boldsymbol{e}_j\|_2}{\sum_{j=1}^{i_k}|\alpha_j|\|\boldsymbol{J_F}\boldsymbol{e}_1\|_2} \leq 1, j = 1, \cdots, i_k$, and

$$\frac{1}{L}\log\|\boldsymbol{J_F}\boldsymbol{v}\|_2 \leq \frac{1}{L}\log\sum_{j=1}^{i_k}|\alpha_j|\|\boldsymbol{J_F}\boldsymbol{e}_j\|_2 \tag{A61}$$

$$= \frac{1}{L}\log\left(\frac{|\alpha_1|\|\boldsymbol{J_F}\boldsymbol{e}_1\|_2}{\sum_{j=1}^{i_k}|\alpha_j|\|\boldsymbol{J_F}\boldsymbol{e}_1\|_2} + \sum_{j=2}^{i_k}\frac{|\alpha_j|\|\boldsymbol{J_F}\boldsymbol{e}_j\|_2}{\sum_{j=1}^{i_k}|\alpha_j|\|\boldsymbol{J_F}\boldsymbol{e}_1\|_2}\right)(\sum_{j=1}^{i_k}|\alpha_j|\|\boldsymbol{J_F}\boldsymbol{e}_1\|_2) \tag{A62}$$

$$= \frac{1}{L}\log\left(\frac{|\alpha_1|\|\boldsymbol{J_F}\boldsymbol{e}_1\|_2}{\sum_{j=1}^{i_k}|\alpha_j|\|\boldsymbol{J_F}\boldsymbol{e}_1\|_2} + \sum_{j=2}^{i_k}\frac{|\alpha_j|\|\boldsymbol{J_F}\boldsymbol{e}_j\|_2}{\sum_{j=1}^{i_k}|\alpha_j|\|\boldsymbol{J_F}\boldsymbol{e}_1\|_2}\right) + \frac{1}{L}\log\sum_{j=1}^{i_k}|\alpha_j|\|\boldsymbol{J_F}\boldsymbol{e}_1\|_2 \tag{A63}$$

$$\leq \frac{1}{L}\log\left(1 + \sum_{j=2}^{i_k}\frac{|\alpha_j|\|\boldsymbol{J_F}\boldsymbol{e}_j\|_2}{\sum_{j=1}^{i_k}|\alpha_j|\|\boldsymbol{J_F}\boldsymbol{e}_1\|_2}\right) + \frac{1}{L}\log\sum_{j=1}^{i_k}|\alpha_j|\|\boldsymbol{J_F}\boldsymbol{e}_1\|_2 \tag{A64}$$

$$\leq \frac{1}{L}\sum_{j=2}^{i_k}\frac{|\alpha_j|\|\boldsymbol{J_F}\boldsymbol{e}_j\|_2}{\sum_{j=1}^{i_k}|\alpha_j|\|\boldsymbol{J_F}\boldsymbol{e}_1\|_2} + \frac{1}{L}\log\sum_{j=1}^{i_k}|\alpha_j|\|\boldsymbol{J_F}\boldsymbol{e}_1\|_2 \tag{A65}$$

$$\leq \frac{1}{L}(i_k - 1) + \frac{1}{L}\log\sum_{j=1}^{i_k}|\alpha_j| + \frac{1}{L}\log\|\boldsymbol{J_F}\boldsymbol{e}_1\|_2 \tag{A66}$$

$$\leq \frac{1}{L}(i_k - 1) + \frac{1}{2L}\log(1^2 + \cdots + 1^2)(\alpha_1^2 + \cdots + \alpha_{i_k}^2) + \frac{1}{L}\log\|\boldsymbol{J_F}\boldsymbol{e}_1\|_2 \tag{A67}$$

$$= \frac{1}{L}(i_k - 1) + \frac{1}{2L}\log i_k + \frac{1}{L}\log\|\boldsymbol{J_F}\boldsymbol{e}_1\|_2 \tag{A68}$$

$$\leq \frac{1}{L}(i_k - 1) + \frac{1}{2L}\log i_k + \lambda_i + \frac{\epsilon}{2}. \tag{A69}$$

Thus, if we set $N_0 = \max\{N, \frac{2i_k - 2 + \log i_k}{\epsilon}\}$, then when $L > N_0$ we have $\frac{1}{L}(i_k - 1) + \frac{1}{2L}\log i_k < \frac{\epsilon}{2}$ and

$$\frac{1}{L}\log\|\boldsymbol{J_F}\boldsymbol{v}\|_2 \leq \lambda_i + \epsilon, \ \forall \boldsymbol{v} \in \mathcal{L}_{i-1}\backslash\mathcal{L}_i, \ \|\boldsymbol{v}\|_2 = 1. \tag{A70}$$

Combining Eqs. (A60) and (A70), we have for any $\epsilon > 0$, there is $N_0 \in \mathbb{N}$, such that when $L > N_0$, we always have

$$\lambda_i - \frac{\epsilon}{2} \leq \lambda_i^L = \sup_{\|\boldsymbol{v}_0\|_2=1, \boldsymbol{v}_0\in\mathcal{L}_{i-1}\backslash\mathcal{L}_i}\frac{1}{L}\log\|\boldsymbol{J_F}\boldsymbol{v}_0\|_2 = \frac{1}{L}\log\|\boldsymbol{J_F}\|_{2,i} \leq \lambda_i + \epsilon. \tag{A71}$$

Thus, if the Lyapunov exponents exist, *i.e.*, the existence of limits of Eq. (A53) , we have

$$\lambda_i = \lim_{L\to\infty}\frac{1}{L}\log\|\boldsymbol{J_F}\|_{2,i} = \lim_{L\to\infty}\lambda_i^L. \tag{A72}$$

### A.6.2 Singular value distributions of Jacobian matrices of deep function coupling

In Sec. A.6.1 we have proved that the Lyapunov exponents (if they exist) are limits of logarithms of subspace spectral norms divided by $L$. Here we use this property to prove the deficiency of numerical ranks, *i.e.*, Eq. (9).

We first introduce the Courant-Fischer min-max theorem [26] of sigular values.

**Theorem 9** (Courant-Fischer Min-max Theorem). *Let $\boldsymbol{A}$ be a $d \times n$ complex matrix and $\sigma_i(\boldsymbol{A})$ denote its $i$-th largest singular value, $i = 1, \cdots, \min\{d, n\}$. Then we have*

$$\sigma_i(\boldsymbol{A}) = \sup_{\dim(\mathcal{V})=i}\inf_{\boldsymbol{v}\in\mathcal{V},\|\boldsymbol{v}\|_2=1}\|\boldsymbol{A}\boldsymbol{v}\|_2, \tag{A73}$$

$$\sigma_i(\boldsymbol{A}) = \inf_{\dim(\mathcal{V})=n-i+1}\sup_{\boldsymbol{v}\in\mathcal{V},\|\boldsymbol{v}\|_2=1}\|\boldsymbol{A}\boldsymbol{v}\|_2, \tag{A74}$$

$$\tag{A75}$$

*where $\mathcal{V}$ traverses subspaces of $\mathbb{R}^n$.*

This theorem also serves as one of the definitions to singular values.

Now assume that $\dim(\mathcal{L}_0\backslash\mathcal{L}_1) = r$, and we consider only the case of $d = n$ for simplicity. For any $\epsilon > 0$, we have $N \in \mathbb{N}$ such that when $L > N$,

$$\inf_{\boldsymbol{v}\in\mathcal{L}_0\backslash\mathcal{L}_1, \|\boldsymbol{v}\|_2=1} \|\boldsymbol{J_F}\boldsymbol{v}\|_2 \geq \exp L(\lambda_1 - \epsilon) \tag{A76}$$

$$\tag{A77}$$

due to Eq. (A55). As $\dim(\mathcal{L}_0\backslash\mathcal{L}_1) = r$, by Theorem 9, we have

$$\sigma_1(\boldsymbol{J_F}) \geq \cdots \geq \sigma_r(\boldsymbol{J_F}) \geq \exp L(\lambda_1 - \epsilon). \tag{A78}$$

For $\boldsymbol{v} \in \mathcal{L}_0 = \mathbb{R}^n$ and $\|\boldsymbol{v}\|_2 = 1$, as $\mathcal{L}_0 = \mathcal{L}_0\backslash\mathcal{L}_1 \oplus \cdots \oplus \mathcal{L}_h\backslash\mathcal{L}_{h+1}$ ($\oplus$ denotes direct sum of linear quotient subspaces in the Banach space), there is $\boldsymbol{v}_i \in \mathcal{L}_{i-1}\backslash\mathcal{L}_i, i = 1, \cdots, h+1$, such that

$$\|\boldsymbol{v}_1\|_2^2 + \cdots + \|\boldsymbol{v}_{h+1}\|_2^2 = 1 \tag{A79}$$

and

$$\boldsymbol{v} = \boldsymbol{v}_1 + \cdots + \boldsymbol{v}_{h+1}. \tag{A80}$$

Then by the conclusion of Sec. A.6.1, we have

$$\limsup_{L\to\infty} \|\boldsymbol{J_F}\boldsymbol{v}\|_2 \leq \limsup_{L\to\infty} \|\boldsymbol{J_F}\boldsymbol{v}_1\|_2 + \cdots + \limsup_{L\to\infty} \|\boldsymbol{J_F}\boldsymbol{v}_{h+1}\|_2 \tag{A81}$$

$$\leq \|\boldsymbol{v}_1\|_2 \lim_{L\to\infty} \|\boldsymbol{J_F}\|_{2,2} + \cdots + \|\boldsymbol{v}_{h+1}\|_2 \lim_{L\to\infty} \|\boldsymbol{J_F}\|_{2,h+1} \leq \sum_{i=1}^{h+1} \|\boldsymbol{v}_i\|_2 \lambda_i \leq \lambda_1. \tag{A82}$$

Thus there is $N_1 \in \mathbb{N}$, such that when $L > N_1$, we have

$$\sup_{\boldsymbol{v}\in\mathcal{L}_0, \|\boldsymbol{v}\|_2=1} \|\boldsymbol{J_F}\boldsymbol{v}\|_2 \leq \lambda_1 + \epsilon. \tag{A83}$$

As $\dim(\mathcal{L}_1) = n - 1 + 1$, by Theorem 9, we have when $L > N_1$,

$$\sigma_r(\boldsymbol{J_F}) \leq \cdots \leq \sigma_1(\boldsymbol{J_F}) \leq \exp L(\lambda_1 + \epsilon). \tag{A84}$$

In conclusion, when $L > N_0 = \max\{N, N_1\}$, we have

$$\exp L(\lambda_1 - \epsilon) \leq \sigma_1(\boldsymbol{J_F}) \leq \exp L(\lambda_1 + \epsilon). \tag{A85}$$

Thus when $L \to \infty$, we have

$$\sigma_1(\boldsymbol{J_F}) \sim \exp L\lambda_1. \tag{A86}$$

Using the same argument for $\sigma_2(\boldsymbol{J_F}), \cdots, \sigma_n(\boldsymbol{J_F})$, we can find that if let $\hat{\lambda}_1 \geq \hat{\lambda}_2 \cdots \geq \hat{\lambda}_n$ be the Lyapunov exponents counting repetitions, *i.e.*,

$$\hat{\lambda}_k = \lambda_i, \text{ if } \sum_{j=1}^{i-1}\dim(\mathcal{L}_{j-1}\backslash\mathcal{L}_j) < k \leq \sum_{j=1}^{i}\dim(\mathcal{L}_{j-1}\backslash\mathcal{L}_j), i = 1, \cdots, h+1, \tag{A87}$$

then

$$\sigma_i(\boldsymbol{J_F}) \sim \exp L\hat{\lambda}_i. \tag{A88}$$

Note that

$$\hat{\lambda}_1 = \cdots = \hat{\lambda}_r = \lambda_1, \hat{\lambda}_i \leq \lambda_2 < \lambda_1, i = r+1, \cdots, n. \tag{A89}$$

Thus we have

$$\frac{\sigma_i(\boldsymbol{J_F})}{\sigma_1(\boldsymbol{J_F})} \sim \exp L(\hat{\lambda}_i - \hat{\lambda}_1) \to 0, i = r+1, \cdots, n. \tag{A90}$$

As a consequence, $\text{Rank}_\epsilon(\boldsymbol{F}) \leq r$ for any $\epsilon > 0$ when $L \to \infty$.

### A.6.3  Existence of Lyapunov exponents for Jacobian matrices of deep function coupling

In above analysis, we have proven Theorem 4 under the existence of Lyapunov exponents. In this section, we introduce the classical result of multiplicatve ergodic theorem in the specific domain of random matrices, which is proposed by Furstenberg and Kesten [19, 20].

**Theorem 10** (Multiplicatve Ergodic Theorem (Theorem 3.9 of [20])). *Let $\mu$ be a probability measure on all convertible matrices of $\mathbb{R}^{n \times n}$ which satisfies*

$$\mathbb{E}_\mu[\max\{\log \|\boldsymbol{J}_{\boldsymbol{f}^k}^{\pm 1}\|_2, 0\}] < \infty, \ k = 1, \cdots, L. \tag{A91}$$

*If each $\boldsymbol{J}_{\boldsymbol{f}^k}$ independently follows $\mu$, then we have a chain of subspaces $\{0\} = \mathcal{L}_{h+1} \subset \mathcal{L}_h \subset \cdots \subset \mathcal{L}_1 \subset \mathcal{L}_0 = \mathbb{R}^n$ and corresponding postive real constants $\lambda_1 > \lambda_2 > \cdots > \lambda_{h+1}$ such that almost surely*

$$\lambda_i = \lim \frac{1}{t} \log \|\boldsymbol{J}_{\boldsymbol{F}}\boldsymbol{v}\|_2, \ \forall \boldsymbol{v} \in \mathcal{L}_{i-1} \backslash \mathcal{L}_i, \ i = 1, \cdots, h + 1, \tag{A92}$$

*which means the existence of the Lyapunov exponents.*

Combining this theorem and the arguments above, we can finally prove Theorem 4.

## A.7  Proof to Theorem 5

This theorem can be deduced from the Lyapunov components of Ginibre matrices (polynomial ensemble of square matrices sampled *i.i.d* from standard Gaussian).

**Theorem 11** (Exact Lyapunov Exponent Distribution for Ginibre Matrices [43]). *If $\mu$ in Theorem 10 is standard Gaussian, then $h + 1 = n$, and*

$$\lambda_i = \log\left(2 + \psi(\frac{n - i + 1}{2})\right), i = 1, \cdots, n. \tag{A93}$$

Combining this theorem with Theorem 4 can directly yield our result.

## A.8  Proof to Corollary 1

This theorem is the direct result of Theorems 4 and 11. Note that it is easy to get

$$\lambda_1 = \lim_{L \to \infty} \frac{1}{L} \log \|\boldsymbol{J}_{\boldsymbol{F}}\|_2 \tag{A94}$$

for standard Gaussian $\mu$.

# B  Influences of Structures

**Skip Connection**  Skip Connection is the most direct method to solve rank diminishing at the initialization period. In our formulation, the definition of a layer network requires it to accept inputs purely from its predecessor layer as

$$\boldsymbol{x}^i = \boldsymbol{f}^i(\boldsymbol{x}^{i-1}), \ \boldsymbol{x}^{i-1} = \boldsymbol{f}^{i-1}(\boldsymbol{x}^{i-2}). \tag{A95}$$

However, when we add a skip connection from its ancestor layer $\boldsymbol{f}^s, s < i - 1$, we have

$$\boldsymbol{x}^i = \boldsymbol{f}^i(\boldsymbol{x}^{i-1}, \boldsymbol{x}^s), \boldsymbol{x}^{i-1} = \boldsymbol{f}^{i-1}(\boldsymbol{x}^{i-2}), \boldsymbol{x}^{i-2} = \boldsymbol{f}^{i-2} \circ \cdots \circ \boldsymbol{f}^s(\boldsymbol{x}^{s-1}), \boldsymbol{x}^s = \boldsymbol{f}^s(\boldsymbol{x}^{s-1}). \tag{A96}$$

It actually makes the coupling of layers

$$\hat{\boldsymbol{f}}^s = \left(\begin{array}{c} \boldsymbol{f}^{i-1} \circ \cdots \circ \boldsymbol{f}^s \\ \boldsymbol{f}^s \end{array}\right), \tag{A97}$$

the true predecessor layer to $\boldsymbol{f}^i$, as

$$\boldsymbol{x}^i = \boldsymbol{f}^i(\hat{\boldsymbol{x}}), \ \hat{\boldsymbol{x}} = \hat{f}^s(\boldsymbol{x}^{s-1}). \tag{A98}$$

Thus the true layer depth is cut down by $i - s$, remaining $L - (i - s)$ layers. Skip connection is usually used with the residual network. This structure can ease rank diminishing inside the layer $\hat{f}^s$, which we will discuss later. Overall, skip connection shortens the length of the chain of Jacobian matrices, thus restraining rank diminishing.

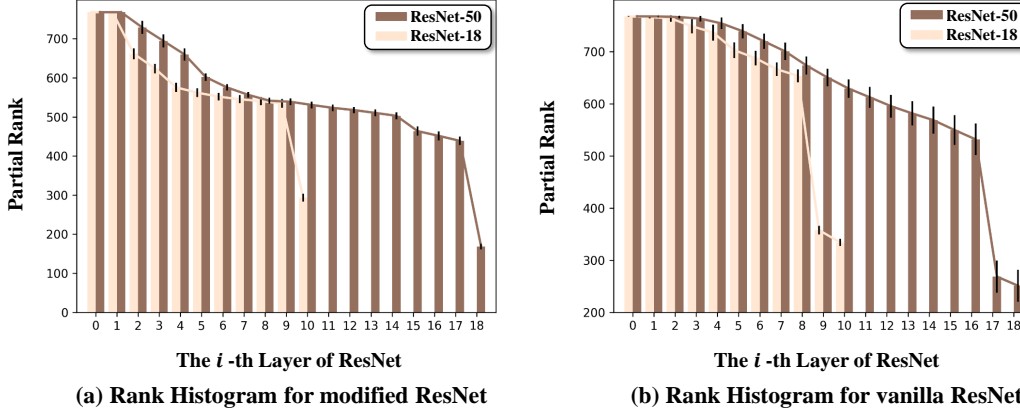

(a) Rank Histogram for modified ResNet    (b) Rank Histogram for vanilla ResNet

Figure A5: The partial rank of Jacobian matrices and perturbed PCA dimensions at the $i$-th layer of the modified ResNet-18 and ResNet-50 on ImageNet. We remove the operation of downsampling so that the feature dimension of the modified ResNet does not change (*e.g.*, $H \times W \times C = 802,816$ for ResNet-50). (a) The results of the modified ResNet-18 and ResNet-50. (b) The results of the initial ResNet-18 and ResNet-50. All the results are measured before training and using random initialization.

**BatchNorm**   Some previous works [13, 5] discuss the role of BatchNorm in restraining rank diminishing. They show that BatchNorm may slow down the speed of rank diminishing in neural networks in some specific cases.

**Residual Network**   Residual Network is another useful tool to restrain rank diminishing at the initialization period. The residual network $r$ has the form

$$\boldsymbol{x}^o = \boldsymbol{r}(\boldsymbol{x}^i) = \boldsymbol{x}^i + \mathrm{Res}(\boldsymbol{x}^i), \tag{A99}$$

where $\boldsymbol{x}^o$ and $\boldsymbol{x}^i$ are the output feature and input feature, respectively. Usually the residual term $\mathrm{Res}(\boldsymbol{x}^i)$ is small compared with the input $\boldsymbol{x}^i$ at the initialization period, as is pointed out by related works [22]. Assume that

$$\|\boldsymbol{J}_{\mathrm{Res}}\|_2 < \epsilon, \tag{A100}$$

where $\epsilon$ is very small. Then we have

$$\boldsymbol{J_r} = \boldsymbol{I} + \boldsymbol{J}_{\mathrm{Res}} \tag{A101}$$

is a diagonally dominant matrix, thus it has full rank. This means its kernel space $\mathrm{Ker}(\boldsymbol{J_r}) = \{0\}$ is a zero dimension space. Thus by the Rank Theorem, for any predecessor layer $\boldsymbol{f}$, $\boldsymbol{r} \circ \boldsymbol{f}$ will not lose rank as

$$\begin{aligned} \mathrm{Rank}(\boldsymbol{r} \circ \boldsymbol{f}) = \mathrm{Rank}(\boldsymbol{J_r}\boldsymbol{J_f}) &= \mathrm{Rank}(\boldsymbol{J_f}) - \dim(\mathrm{Ker}(\boldsymbol{J_r}) \cap \mathrm{Im}(\boldsymbol{J_f})) \\ &= \mathrm{Rank}(\boldsymbol{J_f}) = \mathrm{Rank}(\boldsymbol{f}). \end{aligned} \tag{A102}$$

However, in the well-trained ResNet18 and ResNet50 networks, we still observe considerable diminishing of ranks in Fig. 1. The reason for this phenomenon could be that, during training the magnitude of the residual term $\mathrm{Res}(\boldsymbol{x})$ becomes large. Then the argument above no longer stand and hence the rank becomes lower. Taking the 16-th layer of the ResNet50 as example, we find that the rank of this layer drops from 530 to 119 after training, while the relative magnitude $\frac{\|\mathrm{Res}(\boldsymbol{x})\|_2}{\|\boldsymbol{x}\|_2}$ increases from 0.5127 to 0.9557 after training. This may explain why the residual connection is less effective in preventing network ranks after training.

**Influences of the Pooling Layers and Width**   To better validate the rank behavior of deep neural networks (*e.g.*, ResNet), we remove the operation of downsampling in the ResNet so that the feature dimension (*e.g.*, $x \in \mathbb{R}^{H \times W \times C=802,816}$ for ResNet-50) will not change. This modified ResNet can exclude the effect of pooling layers and changes of layer width. As shown in Fig.A5, we show the partial rank of Jacobian matrices and perturbed PCA dimensions at the $i$-th layer of the modified ResNet-18 and ResNet-50 on ImageNet. We can find that the curves of partial ranks and

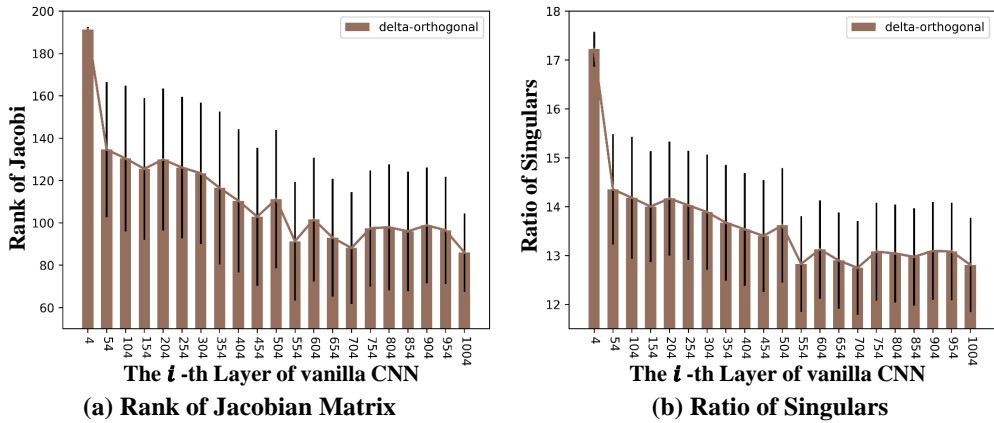

| (a) Rank of Jacobian Matrix | (b) Ratio of Singulars |

Figure A6: The rank of Jacobi matrices and perturbed PCA dimensions at the $i$-th layer of the 10,000 CNN initialized by Delta-Orthogonal on CIFAR-10.

perturbed PCA dimensions share a similar and consistent trend of decreasing as the initial networks. The consistent behavior of partial ranks and perturbed PCA dimensions also shows a monotonic decreasing property of network ranks. Thus the overall trend of rank diminishing seems to be independent of the pooling layers and changes of width. On the other hand, the partial rank witnesses a considerable drop near the terminal layer after applying pooling layers, which means it does have an effect on the network ranks.

## C  Orthogonal Initialization

Previous work [62] has demonstrated that under carefully designed initialization rule, we can train very deep (1,0000 layer) plain CNNs. It is then curious to investigate that whether the phenomenon of rank diminishing happens in this case. To this end, we empirically measure the numerical rank of Jacobi matrices and perturbed PCA dimensions of an extremely deep CNN initialized by the Delta-Orthogonal method [62]. While computing the Jacobi matrices of very deep networks is infeasible in time, we only compute the 4-1,004 layer of the network. We omit the first 4 layers as there are downsampling architectures. We measure two metrics of each layer:

1. the numerical rank of the Jacobi matrix;

2. the ratio between sum of small singular values and large singular values $\frac{\sum_{i=11}^{3072} \sigma_i}{\sum_{i=1}^{10} \sigma_i}$, where 3,072 is the width of this network, and $\sigma_1 \geq \sigma_2 \geq \cdots \geq \sigma_{3072}$ are singular values of the Jacobi matrix.

As shown in Fig A6, a generally decreasing trend of the Jacobi rank and singular ratio can be observed. This result is consistent with Theorem 4. Although orthogonal initialization can suppress rank decrease to a certain extent, the impact of low rank cannot be ignored when the network is deep enough.

## D  Code

Algorithm A1 provides the pseudo-code of partial rank of the Jacobian. The implementation of the Algorithm A1 can refer to the 'rank_jacobian.py' python file.

Algorithm A2 provides the pseudo-code of perturbed PCA dimension of feature spaces. The implementation of the Algorithm A2 can refer to the 'rank_perturb.py' python file.

Algorithm A3 provides the pseudo-code of the classification dimension. The implementation of the Algorithm A3 can refer to the 'run_cls_dim.py' python file.

Algorithm A4 provides the pseudo-code of independence deficit. The implementation of the Algorithm A4 can refer to the 'run_deficit.py' python file.

| Arch. | Network | Activ. | #Param. | Main Block | #Layer | Top-1 Acc. |
|---|---|---|---|---|---|---|
| ResNets | ResNet-18 [23] | ReLU [41] | 11.7M | Bottleneck | 11 | 69.8% |
| | ResNet-50 [23] | ReLU [41] | 25.6M | Bottleneck | 19 | 76.1% |
| MLP-like | GluMixer-24 [49] | SiLU [24] | 25.0M | Mixer-Block | 24 | 78.1% |
| | ResMLP-S24 [53] | GELU [24] | 30.0M | Mixer-Block | 24 | 79.4% |
| Transformer | ViT-T [16] | GELU [24] | 5.7M | ViT-Block | 13 | 75.5% |
| | Swin-T [37] | GELU [24] | 29.0M | Swin-Block | 18 | 81.3% |

Table A2: Information of networks used in empirical validations. All pretrained on ImageNet.

## E   Experiments Setup

Information of those networks used in validations is listed in Tab. A2. When measuring rank, we set $\epsilon = \text{eps} \times N$, where eps is the digital accuracy of $\text{float}32$ (*i.e.*, $1.19e - 7$) and $N$ is the number of singular values of the matrix to measure. This threshold represents the minimum digital accuracy of numerical rank we can capture in data stored as $\text{float}32$. All the experiments are conducted on the validation set of ImageNet and NVIDIA A100-SXM-80G GPUs.

## F   Partial Rank of the Jacobian: Estimating Lower Bound of Lost Rank in Deep Networks

To enable the validation of trend of the network ranks, we propose to compute only the rank of sub-matrices of the Jacobian as an alternative. Those sub-matrices are also the Jacobian matrices with respect to a fixed small patch of inputs. Rigorously, given a function $\boldsymbol{f}$ and its Jacobian $\boldsymbol{J_f}$, we denote partial rank of the Jacobian as the rank of a sub-matrix of the Jacobian that consists of the $j_1$-th, $j_2$-th,...,$j_K$-th column of the original Jacobian

$$\text{PartialRank}(\boldsymbol{J_f}) = \text{Rank}(\text{Sub}(\boldsymbol{J_f}, j_1, ..., j_K)) = \text{Rank}((\partial \boldsymbol{f}_i / \partial \boldsymbol{x}_{j_k})_{d \times K}), \qquad (A103)$$

where $1 \leq j_1 < \ldots < j_K \leq n$. We can efficiently compute sub-matrix of the Jacobian by zero padding to small patches of input images. For any data point $\boldsymbol{x} \in \mathbb{R}^n$, let $\text{Sub}(\boldsymbol{x}, j_1, ..., j_K) = (\boldsymbol{x}_{j_1}, ..., \boldsymbol{x}_{j_K})^T \in \mathbb{R}^K$, and $\boldsymbol{\psi}$ pad $\text{Sub}(\boldsymbol{x}, j_1, ..., j_K)$ to the spatial size of $\boldsymbol{x}$ with zeros: $\boldsymbol{\psi}(\text{Sub}(\boldsymbol{x}, j_1, ..., j_K)) = (0, .., 0, \boldsymbol{x}_{j_1}, 0, ..., \boldsymbol{x}_{j_K}, 0, ..., 0)^T \in \mathbb{R}^n$ with $\boldsymbol{\psi}(\text{Sub}(\boldsymbol{x}, j_1, ..., j_K))_{j_k} = \boldsymbol{x}_{j_k}, k = 1, ..., K$. We then have $\boldsymbol{J_{f \circ \psi}} = \text{Sub}(\boldsymbol{J_f}, j_1, ..., j_K)$. As $K$ can be very small compared with $n$, computing $\boldsymbol{J_{f \circ \psi}}$ can be very cheap in time and space. The partial rank of Jacobian matrices of the network layers measures information captured among the spatial footprint $j_1, ..., j_K$ of the original input. They inherit the order relation of the rank of full Jacobian matrices. Thus we can validate the rank diminishing of network Jacobian matrices through the partial rank.

**Lemma 2.** *For differentiable $\boldsymbol{f}_1, \boldsymbol{f}_2$, $|\text{Rank}(\boldsymbol{f}_1) - \text{Rank}(\boldsymbol{f}_2 \circ \boldsymbol{f}_1)| \geq |\text{Rank}(\text{Sub}(\boldsymbol{f}_1, j_1, \ldots, j_K)) - \text{Rank}(\text{Sub}(\boldsymbol{f}_2 \circ \boldsymbol{f}_1, j_1, \ldots, j_K))|, \forall 1 \leq K \leq n, 1 \leq j_1, \ldots, j_K \leq n$. Thus variance of partial ranks of adjacent sub-networks gives a lower bound on the variance of their ranks.*

### F.1   Partial Rank of Jacobians under Different Input Patches

In Fig. A7 we report partial ranks of different input image patches (marked with colored boxes in Fig. A7(a)) for the layers of ResNet-50 on ImageNet. We can find that the curves of partial ranks share a similar and consistent trend among different input patches. Thus, picking one patch, for example, the central patch of $16 \times 16 \times 3$ pixels we use in Sec. 6.1, could be enough to demonstrate the overall behavior of network ranks. The consistent behavior of all those partial ranks also shows that partial rank is a good tool to investigate network ranks.

## G   Estimating Dimension Diminishing in Features

Measuring the intrinsic dimension of feature manifolds is known to be hard. However, we manage to give a rough estimation to the dimension dropped by different layer networks. To do this, we use a new metric called the Perturbed PCA Dimension. It measures the expectation of PCA dimension of small local neighborhoods over the feature manifold.

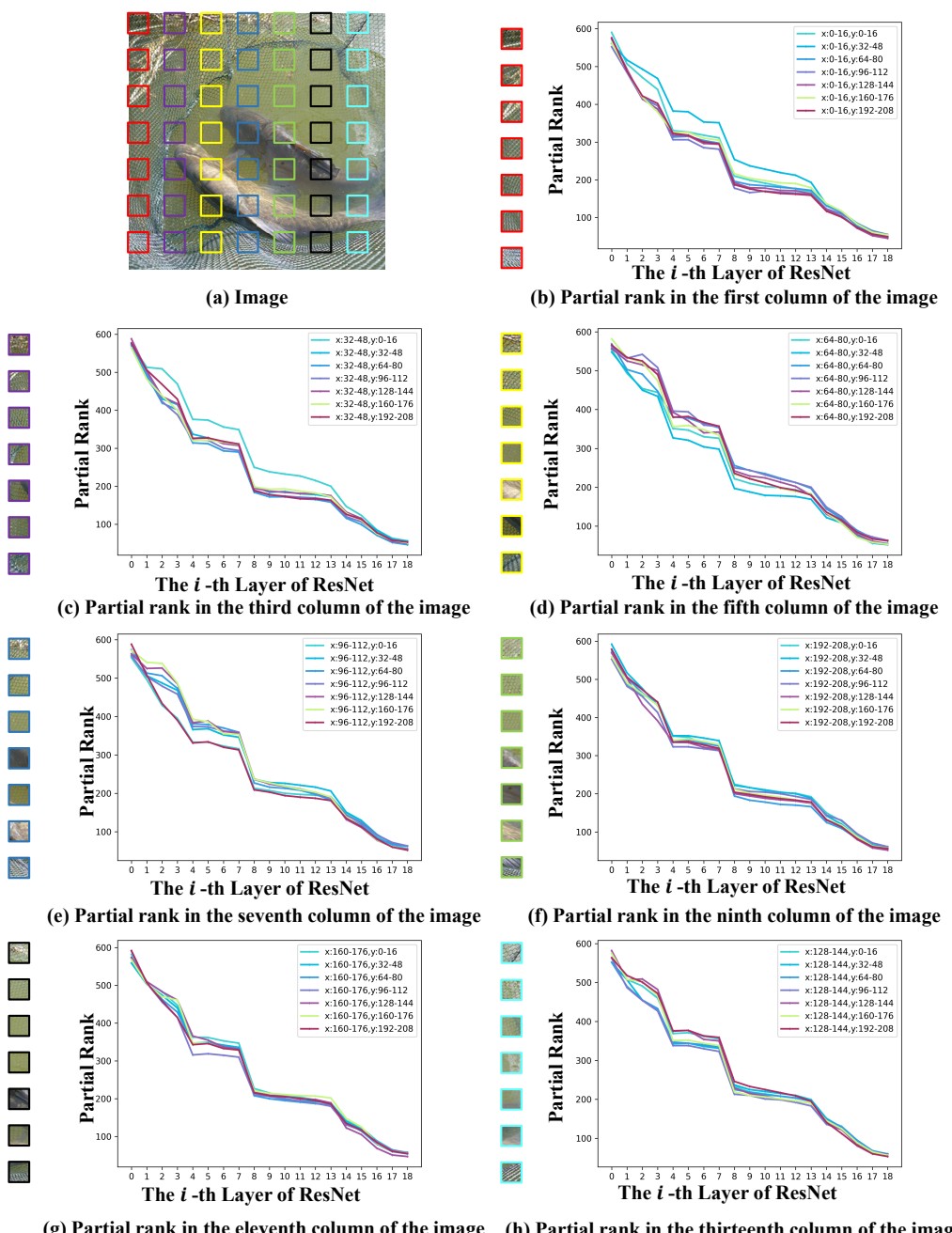

Figure A7: The partial ranks of different input patches at the $i$-th layer of ResNet-50 on ImageNet.

Let $\boldsymbol{F}_k$ be the $k$-th sub-network of the whole network $\boldsymbol{F}$. We want to measure the Perturbed PCA Dimension of $\boldsymbol{F}_k(\mathcal{X})$, where $\mathcal{X}$ is the input data domain. To this end, we compute

$$\text{PertDim} = \mathbb{E}_{\boldsymbol{x}\sim\mathbb{P}_\mathcal{X}}[\text{PCADim}(\{\boldsymbol{F}_k(\boldsymbol{x}+\boldsymbol{\epsilon}) : \boldsymbol{\epsilon}\sim\mathcal{N}(\mathcal{O},\delta\mathcal{I})\})], \tag{A104}$$

where PCADim for a set is the number of PCA eigenvalues larger than a threshold $\xi$. For each point $\boldsymbol{x}$, we sample 50,000 different perturbation $\boldsymbol{\epsilon}$ to compute the PCA dimension of the neighborhood of $\boldsymbol{F}_k(\boldsymbol{x})$. When computing the PCA dimension, we set $\delta = 1e{-}3$ and $\xi = 1.19e{-}7\times 50000\times\text{eig}_{\max}$, where $\text{eig}_{\max}$ is the largest PCA eigenvalue. We then compute the mean value of PCA dimensions over the neighborhood of 100 random samples in the validation set of ImageNet as the final result.

We do not use PCA dimension of the feature manifolds directly as it is unable to cope with the highly non-linear structure of intermediate feature manifolds. However, the Perturbed PCA Dimension is able to estimate the dimensions of local neighborhoods of points in the feature manifolds. As local neighborhoods can be viewed as linear if the network is smooth, the Perturbed PCA Dimension could be more feasible than PCA dimension in our case. We provide the pseudo-code to compute the Perturbed PCA Dimension in Algorithm A2.

However, the perturbation is made in the ambient space of the input data manifold $\mathcal{X}$ rather than the data manifold itself. Thus this estimation may considerably overestimate the intrinsic dimensions of feature manifolds. So we merely care about how many Perturbed PCA Dimensions are lost by a sub-network instead of its own Perturbed PCA Dimension. We call this quantity $\Delta$ Dimension, which is the difference between the Perturbed PCA Dimension of the current layer and that of the input layer for the given deep network. As shown in Fig. A8, we show the dropped dimensions of different feature layers of the CNN, MLP, and Transformer architectures on ImageNet. The results show that the Perturbed PCA Dimensions of feature manifolds of most networks decrease as the networks get deeper, thus confirming the rank diminishing principle we propose in Theorem 2.

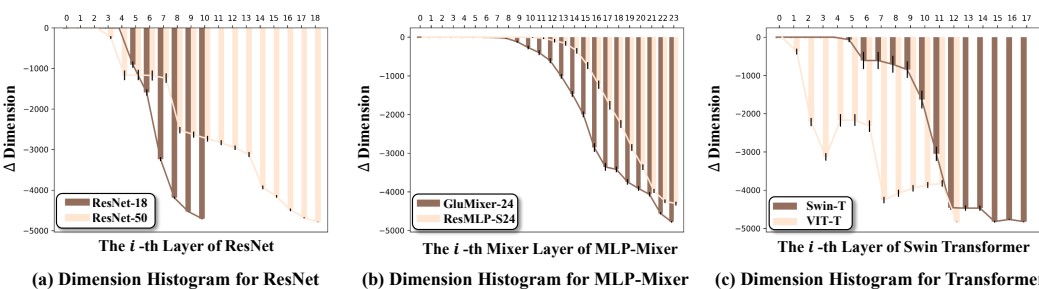

(a) Dimension Histogram for ResNet    (b) Dimension Histogram for MLP-Mixer    (c) Dimension Histogram for Transformer

Figure A8: Dropped Perturbed PCA Dimension of different layers. $\Delta$ Dimension for the $i$-th layer is the difference between the Perturbed PCA Dimension of the $i$-th layer and that of the input layer of the CNN, MLP, and Transformer architectures on ImageNet.

---

**Algorithm A1** Pseudocode of Partial Rank of the Jacobian.

---

```
# image: input images
# model: network
# row_idx, col_idx, patch_size: select a patch of the image to calculate Jacobian matrix

from functools import partial
import torch.nn.functional as functional

def Jacobian_rank(image, model):
    # select a patch of the image to calculate Jacobian matrix
    assert image.size(2) == image.size(3)
    image_size = image.size(2)
    image = image[:, :, row_idx:row_idx + patch_size, col_idx:col_idx + patch_size]
    zero_pad = partial(functional.pad, pad=[(image_size - patch_size) // 2 for _ in range(4)], value=0.)

    # calculate the jacobian matrix
    jacobian_matrix = jacobian(partial(net.forward, preprocess=zero_pad), image)

    # adopt trick to predict the singular values
    jacob = jacob.view(-1, image.size(1) * patch_size * patch_size)
    jacob = matmul(jacob.T, jacob)

    # calculate the partial rank of Jacobian matrix
    return matrix_rank(jacob, symmetric= True)
```

---

`matmul`: matrix multiplication; `jacobian`: calculate the jacobian matrix; `matrix_rank`: calculate the numerical rank of matrix.

---

**Algorithm A2** Pseudocode of Perturbed PCA Dimension of Feature Spaces.

---

```
# image: input images
# model: network
# mag_perturb: magnitude of perturbations
# n_perturb: number of perturbations

def Perturbed_dimension(image, model, mag_perturb=1e-3, n_perturb=5000):
    # extract features with random perturbations
    features = []
    for _ in range(n_perturb):
        # sample random perturbation from Gaussian distribution
        perturb = randn_like(image) * mag_perturb
        # extract feature
        feature = model(image + perturb)
        features.append(feature)
    features = concatenation(features, dim=0)

    # calculate the covariance matrix
    x = input- mean(input, dim=0)
    x = x.view(x.size(0), -1)
    cov_matrix = matmul(x.T, x) # covariance matrix

    # calculate the perturbed PCA dimensions
    return matrix_rank(cov_matrix, symmetric= True)
```

---

`matmul`: matrix multiplication; `randn_like`: sample a random tensor from a Gaussian distribution; `matrix_rank`: calculate the numerical rank of matrix.

**Algorithm A3** Pseudocode of the Classification Dimension of the Final Feature Manifold.

```
# image: input images
# model: network
# target: ground-truth labels
# acc_ratio: threshold for measuring intrinsic dimensions of final features

def PCA(X, n_components):
    n = X.shape[0]
    X_mean = mean(X, dim=0, keepdim=True)
    X = X - X_mean
    covariance_matrix = 1 / n * matmul(X.T, X)
    eigenvalues, eigenvectors = evd(covariance_matrix, eigenvectors=True)
    eigenvalues = norm(eigenvalues, dim=1) # modulus of complex numbers
    idx = argsort(-eigenvalues)
    eigenvectors = eigenvectors[:, idx]
    eigenvectors = eigenvectors[:, :n_components]
    return eigenvectors

def Feature_projection(X, V):
    X_proj = zeros_like(X)
    for component_idx in range(V.size(1)):
        eig_vec = V[:, component_idx].unsqueeze(-1)
        eig_vec_norm = eig_vec / norm(eig_vec, p=2, keepdim=True)
        w_proj = matmul(X, eig_vec_norm)
        X_proj_i = w_proj * eig_vec_norm.T
        X_proj += X_proj_i
    return X_proj

def Intrinsic_dimension(image, model, target, acc_ratio=0.95):
    # pre-extract features and calculate original classification accuracy
    feats = model(image) # [n_samples * n_channels]
    acc_ori = calc_acc(feats, target)

    for n_component in range(1, feats.size(1)):
        # compute the eigenvalues and eigenvectors of a real square matrix
        components = PCA(feats, n_component) # [n_channels * n_component]

        # reconstruct features with principal components
        feats_rec = Feature_projection(feats, components)

        # calculate classification accuracy
        acc = calc_acc(feats_rec, target)

        # return classification dimension
        if acc >= acc_ratio * acc_ori:
            return n_component
```

`matmul`: matrix multiplication; `evd`: eigen value decomposition; `calc_acc`: calculating classification accuracy.

**Algorithm A4** Pseudocode of Independence Deficit.

```
# image: input images
# model: network
# target2index: dictionary mapping from category index to sample indices
# lr: learning rate for Lasso optimization
# n_iteration: number of iterations for Lasso optimization
# w_reg: weight of the L1 regularization term

def Feature_split(feats, class_i, target2index):
    sample_indices = target2index[class_i]
    start_idx, end_idx = sample_indices[0], sample_indices[-1]
    feats_i = feats[start_idx:end_idx+1, :]
    feats_i_n = concatenation((feats_i[:, :class_i], feats_i[:, class_i+1:]), dim=1)
    feats_i_p = feats_i[:, class_i:class_i+1]
    return feats_i_n, feats_i_p

def Independence_deficit(image, model, target2index, lr=1e-5, n_iteration=5000, w_reg=20.0):
    # pre-extract logits
    logits = model(image) # [n_samples * n_classes]

    # Lasso optimization
    for class_i in range(logits.size(1)):
        # split features by category index
        feats_n, feats_p = Feature_split(logits, class_i, target2index)

        # initialize the linear coefficients of category i
        param = Parameter(zeros(feats_n.size(1), 1))

        # start training
        for _ in range(n_iteration):
            loss = mse(matmul(feat_n, param), feat_p) + w_reg * l1_norm(param)
            loss.backward()
            param -= lr * param.grad

        # save trained coefficients
        save(param)
```

matmul: matrix multiplication; mse: mean squared error; l1_norm: sum of the magnitudes of the vectors in a space.