# OpenReview forum: "Rank Diminishing in Deep Neural Networks"
_NeurIPS.cc/2022/Conference — NeurIPS 2022 Accept_

### Official Review · Reviewer_4Q14 · 2022-06-25

**Rating:** 5
**Confidence:** 3
**Soundness:** 2 fair
**Presentation:** 3 good
**Contribution:** 3 good

**Summary:**

This work presents some theoretical results that imply that the rank of the Jacobian between the inputs and features of deep networks is non-increasing with depth. They predict that in some settings it should in fact decrease exponentially with depth to some fixed value.

They also develop efficient methods to estimate the Jacobian rank of real networks and show empirically that it indeed decreases with depth across a number of different architectures.

**Questions:**

Following my remarks in the previous section, how difficult do the authors think it would be to relate some of the predictions in a quantitative way to various aspects of the network architecture?

**Limitations:**

Limitations have been addressed

**Strengths And Weaknesses:**

The effects of depth on the learned representations in deep networks and their geometric structure is an important area of study. While this work contains an interesting combination of theoretical and empirical results, I believe the connection between the two would have to be made more concrete.

The result about non-decreasing rank follows from the basic compositional structure of the network as the authors suggest, yet it is unclear that the rank must decrease. In fact, there is a vast literature on signal propagation in deep networks that approaches this question from a different angle (by studying covariance between hidden features as a function of depth, in which case convergence to certain fixed points should essentially be equivalent to the rank of the representation collapsing [1, 2]). This literature also highlights ways to avoid this phenomenon with a careful choice of initialization, and relies on modeling the dynamics of the correlations as a function of initialization hyperparameters. This allows one for example to train convnets of depth 10000 [2]. In the simplest case of a network with orthogonal weights and no non-linearities, it is clear for example that there is no decrease in rank, so there are clearly ways that it can be avoided.

Another related issue is that the results are vague in the sense that the behavior of the rank is not connected in a quantitative way with the structure of the network (i.e. the choice of nonlinearity, initialization, etc). I think the submission would be much more compelling if the results could take these into account and make predictions about their effects on the rank. For example, how is the rank one converges to or the speed of the rank decay related to properties of the network?

An additional, related concern is the connection between the experiments and the theory. The experiments that attempt to show exponential decay of the rank are not plotted on a logarithmic scale, which makes it hard to understand whether the decay there is indeed exponential or follows some other law. In addition, it appears that the rank decay in the case of resnets may be influenced more by the pooling layers or changes in width than any other operation, yet no mention of this is made in the text.


[1] Poole, Ben, et al. "Exponential expressivity in deep neural networks through transient chaos." Advances in neural information processing systems 29 (2016).

[2] Xiao, Lechao, et al. "Dynamical isometry and a mean field theory of cnns: How to train 10,000-layer vanilla convolutional neural networks." International Conference on Machine Learning. PMLR, 2018.

---

> ### Author Response · Authors · 2022-08-02
> **Thank you for the detailed feedback and questions!**
>
> Thank you for the detailed feedback and questions. Below we address the concerns separately.
>
> ### **Q: It is unclear that the rank must decrease.**
>
> **A:** There could be a misunderstanding. The purpose of this paper is not to prove that rank must decrease in all cases, which is apparently wrong, as pointed out by the reviewer. This paper's purpose is to explain why we usually observe a low-rank network in practice and to provide arguments to support the low-rank network assumption in various domains of deep learning. Numerous methods are derived from an assumption that low-rank structures are to be preferred (like those mentioned in the first paragraph of sec. 1). Yet currently, few works discuss why the low-rank structures are preferred. So we fully agree that rank will not decrease in many carefully manual designed cases. This fact is orthogonal with the topic of this paper.
>
>
> ### **Q: The covariance converges to certain fixed points should essentially be equivalent to the rank of the representation collapsing.**
>
> **A:** Thank you for raising this related direction of research. If the covariance matrix is precisely the fixed point defined in Eq. 2.6 of [2], i.e. $\Sigma_{i,i'}^{\*}=q^*(\delta_{i,i'}+(1-\delta_{i,i'})c^*)$, then indeed the network should be low-rank. However, convergence is another thing. The covariance converging to a fixed point does not necessarily mean that rank of the networks will collapse or not. In fact, we can have a simple counter-example. Let the input be a standard Gaussian distribution, and the network layer $f^n(x)=\frac{1}{n}x$. Then the covariance matrix of the n-th layer will be $\Sigma=\Pi_{i=1}^n \frac{1}{i^2}I$ by the property of Gaussian under linear transformation, where $I$ is the identity matrix and the network Jacobi of the n-th layer will  also be $J_{F_n}=\Pi_{i=1}^n \frac{1}{i}I$. Then we can find $\Vert\Sigma-0\Vert_2\rightarrow0$, meaning the covariance converges to a fixed point $0$, while both the rank and numerical rank of the Jacobian matrix will stay full rank. Generally, the covariance matrix is continuous with respect to its elements, but the rank of Jacobi is not continuous with respect to the elements of the Jacobi. So their convergence cannot be equivalent.
>
> From this viewpoint, the main results of this paper are in fact parallel to the two mentioned works.
>
>
> ### **Q: In the simplest case of a network with orthogonal weights and no non-linearities, it is clear for example that there is no decrease in rank, so there are clearly ways that it can be avoided.**
>
> **A:** It may be surprising to find that even the carefully designed network [2] can have rank diminishing at initialization. We think the reason could be that
> 1) **rank is not continuous with respect to the elements of the matrix;**
> 2) there are unavoidable numerical errors in the network;
> 3) accumulated small errors can be large after massive matrix multiplications and cause the numerical rank to lose.
>
> In fact, from a probability perspective, the rows of the standard Gaussian Jacobi matrix are almost orthogonal to each other as they have zero correlations. However, the numerical rank of this case will still diminish to one when the layers get infinitely deep.
>
> In section C of  Appendix, we measure two metrics for the delta-orthogonal initialized CNN network in [2],
> 1) the numerical rank of the Jacobi;
> 2) the ratio between the sum of non-largest singular values and the largest singular values $\frac{\sum_{i=11}^{3072}\sigma_i}{\sum_{i=1}^{10}\sigma_i}$, where $\sigma_1\geq\sigma_2\cdots\geq\sigma_{3072}$ are singular values of the Jacobian matrix.
>
> We report the results in the first 4-1004 layers (we omit the first 3 layers as they have downsampling operations). We find that both these two metrics diminish as the layer gets deeper. This indicates that the network still has an intention to lose ranks when the layer depth is very large.

---

> > ### Author Response · Authors · 2022-08-02
> > **Response to Reviewer 4Q14 (Part II)**
> >
> > ### **Q: The bottom row of Fig. 1 is not a logarithmic scale.**
> >
> > **A:** Thank you for raising this point! We have changed it to a log scale in the updates.
> >
> > ### **Q: it appears that the rank decay in the case of resnets may be influenced more by the pooling layers or changes in width than any other operation**
> >
> > **A:** We have measured the rank decay in ResNets without pooling layers and with the same width. We can find that the new results still follow the same trend of decreasing. The pooling layer does have a considerable effect near the terminal layer, but the overall trend of rank diminishing seems to be independent of it.  These results can be found in the paragraph **Influences of the Pooling Layers and Width** of Appendix section B.
> >
> > ### **Q: the behavior of the rank is not connected in a quantitative way with the structure of the network.**
> >
> > **A:** The quantitative connection between ranks and network structures is indeed an interesting problem. However, in this paper, we intentionally omit the discussion about specific architectures to better serve the main purpose of this work. The main purpose of this paper is to reveal how the two fundamental ingredients, the chain rule of differential and matrix multiplication, can induce rank diminishing in deep networks.  From this viewpoint, we can provide more general arguments for the low-rank preference for deep networks. So we intentionally abstract away from specific structures, as
> > 1) they may weaken the key argument of how the chain rule of differential and matrix multiplication influence the ranks;
> > 2) the structures for deep networks are numerous; discussions about them will be tedious for 9-page limits;
> > 3) if only discussing the influences of a few structures, then the results cannot be general enough.
> >
> > While in theorems 4 and 5, we still manage to give two quantitative descriptions of the rank behavior of deep networks. We show that the numerical ranks converge to some fixed constants at an exponential speed. This may address the concern about quantitative descriptions of ranks.
> >
> > ### **Q: how difficult do the authors think it would be to relate some of the predictions in a quantitative way to various aspects of the network architecture?**
> >
> > **A:** This is a very good question. The most important thing to note about the rank is that it is not continuous with respect to the elements of the matrix. This makes any analytic analysis of it extremely difficult compared with the covariance matrix. Recent advances in multiplicative ergodic theorem and Ginibre ensembles in random matrix theory provide us with chances to investigate it from a probability perspective.
> >
> > The math community already knows the joint density distribution of singular values of products of (finite) Gaussian matrices, which is a determinantal point process with a correlation kernel that
> > admits a representation in terms of Meijer G-functions. The Jacobian matrix of networks is a smooth function of the weight matrices. Using this function, ideally, we can do integrals to compute the density function of the singular values of the Jacobian matrix. However, for complex structures, we usually cannot get an analytic density. The quantitative predictions for ranks are easy for cases where we can have an analytic density of singular values (an example is a network of the form $f^k(x)=\gamma W_k x+b$), while it could be difficult for the remains.

---

> > > ### Author Response · Authors · 2022-08-08
> > > **We sincerely hope response from the reviewer before the end of the author-reviewer discussion period.**
> > >
> > > Dear Reviewer 4Q14,
> > >
> > > The author-reviewer discussion period is about to end. We want to know whether our previous response can relieve or successfully address any of your initial concerns. If not and you have any further questions, please feel free to share with us so that we can use our last chance to address them. We want to thank you again for your valuable time and efforts. We will be more than happy to receive your response and resolve any further problems.

---

> > > > ### Comment · Reviewer_4Q14 · 2022-08-08
> > > > **Response to authors**
> > > >
> > > > I thank the authors for their clear and detailed response. This has resolved some of the potential misgivings I had regarding the relationship to other work, and I am happy to increase my score.

---

### Official Review · Reviewer_dzFU · 2022-07-07

**Rating:** 7
**Confidence:** 4
**Soundness:** 3 good
**Presentation:** 3 good
**Contribution:** 4 excellent

**Summary:**

The paper studies the dynamics of the rank evolution of the feature maps of a neural network as a function of its depth. By leveraging the abstract definition of rank of a function as the rank of the corresponding Jacobian matrix, the authors can study the rank dynamics in full generality (i.e. without assuming any specific architecture). This results in Theorem 1 (Principle of Rank Diminishing), that finds that the rank of neural network should never increase with depth due to its compositional nature (a neural network can be see as a composition of $L$ functions, where $L$ is the depth). Then, the authors analyze conditions under which the rank strictly diminishes (Theorem 3) and convergence of the rank to specific constants (Theorem 4-5).

Finally, the authors apply their low rank findings to the study of the dependence and correlations between different output classes. They find that the output of some classes of ImageNet (e.g. hamster) can be predicted with a linear combination of the output for irrelevant classes (e.g. broccoli and mouse trap). The authors attribute this problem to the low rank representations of very deep network, as showed by their developed theory.

**Questions:**

1. I would like to ask the authors if in their view there is a way to quantify what they call the "Structural Impetus" for different architectures and normalization layers.

2. Can (and if so, how) their results can be used to design better architectures, initializations, training procedures that better preserve the rank information?

**Limitations:**

I do not see a negative societal impact of this theoretical work.

**Strengths And Weaknesses:**

**Strengths**
1. **Generality and Importance of the Results**: the theoretical results are very general and remarkable, abstracting away from the specific architecture. The only assumption is the compositional nature of the layers, which includes most of architectures but excludes residual networks (as the author mention in the supplementary material).

2. **Paper Organization**: the paper is very clear in explaining the abstract concepts of the first part. Until Theorem 2 (page 4), the theory is easy to digest. At first read, Theorem 1 seems trivial if one thinks about linear networks (i.e. simple product of matrices) and the famous property $\text{rank}(AB) \leq \min(\text{rank}(A), \text{rank}(B))$, but the author do a great job to generalize it to any composition of functions through ideas from topology theory. The other two theorems delve deep into the rank diminishing properties of function compositions, showing an exponential decay of the rank with depth.

3. **Independence Deficit of Feature Manifolds**: Section 5 provides a nice application of the theory, and would probably cause follow up works in trying to understand how one can reduce this undesirable effect of strong dependences between semantically different classes.

**Weaknesses**
1. **Inconsistency of Residual Network**: Skip connections are proposed as a tool to (partially) prevent the rank deficiency problem, and they give a brief theoretical argument in the supplementary material. However, this seems to be in contradiction with Figure 1, where an exponential decay of the rank is observed for ResNets, MLP-Mixers and Transformers, all architectures that adopt skip connections. This could be due to the fact that during training the magnitude of $\text{Res}(x^i)$ becomes large, hence lowering the rank. At initialization, the magnitude of $\text{Res}(x^i)$ can be controlled, e.g. with an appropriate factor inversely proportional to the depth (see for instance [1] for this scaling and [2] for its consequences on the rank). In any case, I found it confusing that skip connections are adopted in almost all the architectures used to exemplify the theory (skip connections that according to the authors should have an opposite effect).

2. (minor) **Presentation Style of Structural and Implicit Impetus**: After brilliantly explaining the principle of rank diminishing, in my view  the concepts of "Structural Impetus" (due to the specific architectural modules) and "Implicit Impetus" (due to the very compositions of infinite modules) of rank diminishing could be better explained. In particular, I would invest some extra lines to better explain why normalization layer prevent rank diminishing, and maybe better introduce some concepts ( or instance "moving along directions" of Theorem 3 is not properly introduced and in general the current version of the Theorem fails to convey a simple and intuitive explanation).

[1] Hanin, Boris, and David Rolnick. "How to start training: The effect of initialization and architecture." Advances in Neural Information Processing Systems 31 (2018).

[2] Noci, Lorenzo, et al. "Signal Propagation in Transformers: Theoretical Perspectives and the Role of Rank Collapse." arXiv preprint arXiv:2206.03126 (2022).

---

> ### Author Response · Authors · 2022-08-02
> **Thank you for the valuable feedback and insightful questions! We are encouraged by your support for this work!**
>
> The authors are grateful for the reviewer's valuable feedback and insightful questions. We are encouraged by your support for this work! Below we address the concerns separately.
>
> ### **Q: Inconsistency of Residual Network.**
>
> **A:** We are grateful for the reviewer pointing out this inconsistency and offering an insightful opinion.
>
> We empirically study this issue in the ResNet50 networks. We find that
>
> 1) The rank of each layer of ResNet50 at initialization is much higher than after training (taking the 16-th layer of ResNet50 as an example, its numerical ranking before and after training is 530 and 119, respectively.);
> 2) The relative magnitude of the Residual term, $\frac{\Vert Res(x^i)\Vert}{\Vert x^i \Vert}$, at initialization is much larger than after training (take the 16-th layer of ResNet50 as an example, the ratio raises from 0.5127 to 0.9557 after training).
>
> This means that the residual connections take effect when they are initialized and have a small magnitude, as is analyzed in Appendix B. While after training, the residual terms become significant and not as effective in preventing rank diminishing. So the reviewer's comment that "this could be due to the fact that during training the magnitude of $Res(x^i)$ becomes large, hence lowering the rank" seems to explain this phenomenon very well.
>
> On the other hand, the skip connection and residual architecture are still effective for deep networks from the following aspects,
>
> 1) it stabilizes the training as in the early period of the training, the rank diminishing can still be eased;
> 2) it will still be better to have them than not, as the residual term is not large enough to totally eliminate the effect of the identity term.
>
> We have revised the content in Appendix B according to the above discussion. Thank you again for raising this point!
>
>
> ### **Q: Why normalization layer prevents rank diminishing?**
>
> **A:** From the perspective of Theorem 4 and 5, the main reason is that normalization techniques can re-normalize the singular value distribution of the feature representations or the networks. For example, the Batch Normalization is motivated to pull the feature representation back to a normal distribution with identical covariance. The option $\frac{x-\mu}{\sigma}$ will make the singular value distribution of the feature covariance matrix more uniform. This can stabilize the rank of the covariance matrix, which is also the dimension of the feature manifolds under regularization assumptions. We have added this discussion in sec. 5 to enhance the connection of the theoretical results with the practice. Thank you for your suggestions!
>
> ### **Q: Intuitive explanation for "Moving along directions in Theorem 3".**
>
> **A:** Thank you for this suggestion. We have added two new examples in sec. 4 below this theorem to illustrate our meaning here. Roughly speaking, this means adding some small perturbations to the input can yield exact zero change in the output.

---

> > ### Author Response · Authors · 2022-08-02
> > **Response to Reviewer dzFU (Part II)**
> >
> > ### **Q: Is there a way to quantify the structural impetus?**
> >
> > **A:** Generally, we think it is very hard to give an exact measure, as it is difficult to sample data along directions in the manifold. However, rough measurements are easy to have. For example, we can do PCA dimension reduction to the input manifold (the data or the input feature manifolds). Then for a given input point, we can add perturbations to its significant PCA components and measure the PCA dimension of the resulting output space. The expectation of decreasing of PCA dimension can be viewed as a rough estimation of the structural impetus. Following this idea, other methods (like [1][2]) that learn the structure of manifolds can also be used to deduce a rough estimation of the structural impetus.
> >
> > [1] A Global Geometric Framework for Nonlinear Dimensionality Reduction, Science
> >
> > [2] The Isomap Algorithm and Topological Stability. Science
> >
> >
> > ### **Q: Can (and if so, how) their results can be used to design better architectures, initializations, and training procedures that better preserve the rank information?**
> >
> > **A:** There are some potential directions to improve the rank information based on our theory. Below we discuss a few, perhaps most straightforward ones, separately.
> >
> > **1) Normalizing the singular value distribution of weight matrices during training.** Previous work on spectral normalization [3] has revealed that normalizing the largest singular value of the weight matrices can help the training of generative models. While according to the theory of numerical rank in this paper, if we can further normalize the top-k largest singular values to be more uniformly distributed, we can preserve the rank of the weight matrices large than $k$. This can hopefully help stabilize the rank of the whole network.
> >
> > **2) Regularizing the residual terms during training.** As is insightfully pointed out by the reviewer, if the magnitude of the residual term becomes too large, it may lower the rank of the network. So a nature thought will be regularizing the residual terms during training. We can regularize the top-k singular value distribution of the residual term or the weights of its CNN blocks as in the above point to make the residual term has higher ranks. Or we can regularize the magnitude of the residual term directly to make the identity term dominate the network rank.
> >
> > **3) Use small width feature layers sparingly and take the necessary dimension for the outputs into account when choosing layer width.** Like pointed out in Theorem 2, in a smooth network, once the rank is lost, it can never get back.  So if the width of some intermediate feature layer is very small, it will lose rank immediately at this layer. Increasing width in the subsequent layers will not bring the lost rank back. Specifically, the transformer network uses a 192-dimensional feature layer near the output layer. Thus the intrinsic dimension of the output manifold will never exceed 192 theoretically. This could be unwise for the classification of massive and diverse categories where a rank of 192 is obviously too small.
> >
> > [3] Spectral Normalization for Generative Adversarial Networks, ICLR, 2018

---

> > > ### Comment · Reviewer_dzFU · 2022-08-08
> > > **Reply**
> > >
> > > I thank the authors for clarifying my concerns and address my curiosities. The additional results on Resnets at initialization vs after training could indeed inspire new directions toward controlling the evolution of the rank during training. Also, more work could be done in future works around the role of normalization layers.
> > >
> > > I agree with Reviewer Gfbz regarding the original ambiguity in the definition for the rank. However, I believe the authors improved their manuscript by updating their definition and discuss its implications.
> > >
> > > Overall, I am still convinced of my score (7), hence advocating for acceptance.

---

### Official Review · Reviewer_Dh79 · 2022-07-08

**Rating:** 6
**Confidence:** 4
**Soundness:** 3 good
**Presentation:** 2 fair
**Contribution:** 2 fair

**Summary:**

This work aims to study the rank of hidden layer representations of neural networks in relation to how deep the layer is in the network. In particular they note that the rank of the hidden layers diminish monotonically as we observe deeper layers. Numerical measures of rank are proposed and motivated. The primary theoretical concerns are the rank of the Jacobian from the input to the i-th layer of the network (essentially a linear approximation of the network mapping to that hidden layer) and the dimension of the feature space for a hidden layer. The paper further investigates the tolerance of the final hidden layer to dimensionality reduction by applying PCA to features space and projecting onto a decreasing number of eigenvectors. The number of eigenvectors remaining when a significant drop in performance is observed from the dimensionality reduction provides an approximation for the intrinsic dimensionality of the hidden layer. Finally, the paper explores the idea that it is possible to use the logits of different categories to classify another category in a dataset. One example is that by merely using -0.923 as a weight on the logit for the "triumphal arch" category it is possible to predict the "junco" category without loss of accuracy.

**Questions:**

No questions as yet.

**Limitations:**

I suggest that the authors be clearer on the conditions required for their theory to help. For example saying "The principle of rank diminishing describes the behavior of general neural networks with almost everywhere smooth components," which does not seem to include ReLU networks but is described as general is unclear.

**Strengths And Weaknesses:**

# Strengths
## Originality
The paper is fairly original with the primary novelty being the rank metrics used and their justification. Additionally the paper touches on some possible connections between symmetry and rank which to my knowledge have not been explored, however, these connections are mainly pointed out but not discussed or treated theoretically.

## Quality
The need for the numerical tools to measure rank is well motivated and the numerical tools themselves makes sense and are justified. The claims that are made appear correct and in-line with the evidence presented.

## Clarity
There is some variance in the clarity of the paper for various sections. The writing is clear and understandable and the mathematical notation is consistent and intuitive which helps the clarity in the earlier sections greatly. Sections 3.2 and 3.3 are examples where the notation made potentially tricky sections more manageable. Figure 4 stands out as a very helpful figure. The effort on that is definitely worth it.

## Significance
The paper touches on some significant points, like the point linking symmetries to lower ranks. The PCA experiment and the experiment on the using categories as predictors for others may be of general interest to the ML community.

# Weaknesses
## Originality
A primary concern of this work is the fact that $Rank(AB) \leq min{Rank(A), Rank(B)}$. This is even mentioned in the paper below equation 8 and is one of the primary tools for the work. This, however, is a well established principal and quite intuitive. Thus, the finding that the rank of the network decreases with layer depth is not surprising. Two possible interesting points: noise increasing network rank and structure avoiding the rank staying the same across layers are mentioned but do not form part of the analysis. The noise aspect is ignored in the theory and removed through the noise tolerant rank measures. The point on monotone decrease over equality of rank due to structure is discussed briefly.

## Quality
The various sections of the paper feel quite loosely connected. Up to Section 4 the work considers whether the rank of the network decreases monotonically. The Section 5 considers PCA just on the final feature space and is the used to point out that low dimensional feature spaces do not hold semantically meaningful features for each category in Section 6. These sections are all related to rank, however, the connections do not seem to go deeper than that. Finally, there are some points where unjustified claims are made (or the phrasing makes these claims appear unjustified). Two examples are "Theorem 5 that investigates the behaviour of all singular values of deep neural networks" when theorem 5 requires hidden layers of the same size and assumes the Jacobians have Gaussian elements (which appears to be unrealistic in its own right) and "The principle of rank diminishing describes the behavior of general neural networks with almost everywhere smooth components" where it is not clear that ReLU networks would even fit this requirement.

## Clarity
Theorem 4 and Theorem 5, which are the most technical aspects of this paper are not given enough space. The clarity of the paper could benefit greatly from a more in-depth treatment of this section. In addition, how the theory of these sections relate to Figure 1 could also be explained more. For example I acknowledge that the shape of the bottom row of Figure 1 is non-linear but to call it exponential (which Theorem 4 and 5 predict) might also be a stretch. Understanding Theorem 4 and 5 would help with interpreting Figure 1. Figure 1 could also use different colours, especially for the bottom row where distinguishing between Jacobian and Feature rank/dimension is not easy. Finally, the notation of Section 5 is not easy to follow, particularly in the meaning of the $i_j$ double subscript where it is not immediately clear what $i$ and $j$ each refer to. Figure 4 does help clarify this a lot and with space constraints fully explaining the new notation may not be feasible.

## Significance
This work appears generally significant, however, its significance is hindered by the same issues noted under the originality section. I feel that this work might spend too much time on the potentially quite obvious points of rank diminishing and on introducing the PartialRank and not enough time on the potentially significant points such as Theorem 4 and 5. My primary recommendation would be to rephrase the work more in line with those theorems.

---

> ### Author Response · Authors · 2022-08-02
> **Thank you for the detailed and in-depth feedback! We have substantially revised the manuscript as suggested.**
>
> The authors are grateful for the detailed and in-depth feedback from Reviewer DH79. We have substantially revised the manuscript as suggested by the reviewer. Below we address the mentioned concerns separately.
>
> ### **Q: Does ReLU fit the requirement of almost everywhere smooth?**
>
> **A:** Yes, ReLU fits this requirement. We use "almost everywhere smooth" to describe functions that have gradients and arbitrary high-order gradients (which means smooth) except for a zero-measure set (which means almost everywhere). Specifically, the ReLU function is non-differentiable only at the zero point, which is a zero-measure set (collections of finitely many points are all zero-measure sets). Apart from the zero point, ReLU is smooth at each point. Thus ReLU function fit the requirements of "almost everywhere smooth."  In fact, most network components are either smooth or smooth except for some isolated points, including ReLu, Pooling, LeakyRelu, Sigmoid, Tanh, SoftMax, Attentions, CNNs, and Dense layers. We have added the exact definition of smooth almost everywhere to the revised version.
>
> ### **Q: The conditions of Theorem 5 appear unrealistic.**
> **A:** We make these assumptions mainly for theoretical convenience and interests. The answer to this question can be divided into two parts, why we assume the fixed size of hidden layers and why we assume Gaussian distributions.
>
> **1) Fixed size of hidden layers:** We make this assumption for the convenience of theoretical analysis of the limiting behavior of infinitely deep networks. Otherwise, some variables of interest may not have limits. Here we want to predict the limit of series $\frac{\sigma_i(J_{F_L})}{\sigma_1(J_{F_L})}$ when the depth L goes to infinity. If we do not use a fixed size for hidden layers, then $\sigma_i(J_{F_L})$ is not well defined, as it may suddenly appear or disappear in the series. In this case, we cannot define the limit $\lim_{L\rightarrow\infty}\frac{\sigma_i(J_{F_L})}{\sigma_1(J_{F_L})}$. In fact, for the same reason, the assumption of fixed hidden layer size is a common practice when analyzing the limiting behavior of infinitely deep networks, as in [1][2][3][4]. Apart from that, some frequently used network architectures, like transformers and RNNs, also satisfy the fixed size assumption.
>
> [1] Deep Equilibrium Models, NeurIPS, 2019
>
> [2] Doubly infinite residual neural networks: a diffusion process approach, JMLR, 2021
>
> [3] Variational Inference for Infinitely Deep Neural Networks, ICML, 2022
>
> [4] Transport analysis of infinitely deep neural network, JMLR, 2019
>
> **2) Gaussian assumptions:** We are interested in this setting as Gaussian is the simplest distribution. Also, according to the central limit theorem, infinitely wide networks tend to have a Gaussian Jacobi matrix after normalization ($\tilde{J_{ij}} =\frac{J_{ij}-\mu_{ij}}{\sigma_{ij}}$). Thus, studying this setting can lead to much stronger results on rank diminishing due to the simplicity of Gaussian. Yet those results are also intuitive for understanding deep networks.
>
> We have added the discussion above in the revised version (in sec.5 above Theorem 4 and Theorem 5).
>
>
> ### **Q: The aspect of noise increasing network rank.**
>
> **A:** The role of noise in the deep network is indeed an interesting problem. However, when considering their role in the rank of networks and features, it follows that a noisy feature manifold almost always has full dimension, and thus a noisy network almost always has full rank.  For small and unwished noise, this will make the discussion about ranks trivial. So we need to remove it in the measuring of numerical ranks. For large and intended noise injected into the network, this property of noise makes it a common practice to increase network robustness and performance like in dropout and other noise injection techniques. In our formulation of numerical ranks, small noise produced by the inaccuracy of the computations can be safely ignored, but large noise that is intended to add will still function as a tool to slow down the decreasing of network ranks. This is contained in the condition of Theorem 1 that only a small noise perturbation (smaller than $\delta_{max}(\epsilon)$) can be removed from the numerical ranks. The role of large noise can also be described by Theorem 4 and 5. If we consider the noise perturbed Jacobi distribution as a new distribution, then they can also satisfy the conditions of those two theorems.

---

> > ### Author Response · Authors · 2022-08-02
> > **Response to Reviewer Dh79 (Part II)**
> >
> > ### **Q: Relations between structures and network ranks.**
> > **Q:** This is really an interesting aspect. But a full discussion of it may be a bit tedious as the structures are quite diverse and numerous in deep networks. So we add a discussion on this topic in Appendix section B, where we discuss the influences of some frequently used architectures on the rank of networks, like skip connections, batch norms, residual networks, and pooling layers.
> >
> >
> > ### **Q: What are the connections between section 4, the final layer PCA results, and the independence deficit results?**
> >
> > **A:** The results of the final layer PCA and independence deficit are to show that **the final layer representations are indeed low-rank** from a numerical perspective and a semantic perspective, correspondingly. This is to support that the diminishing of the ranks is significant for the deep networks, as suggested by Theorem 4 and 5. Without these two results, we may suspect that the diminishing of ranks is modest and can be neglected. The purpose of these two experiments is different from Fig. 1, where we want to directly confirm the **"existence"** of rank diminishing. Here we want to confirm the final **"effect"** of rank diminishing. So they have to be taken place in the terminal feature layers, where the outputs of the network are given. We have reorganized the paper to emphasize this point.
> >
> > For the per-layer feature dimensions, we provide an experiment in sec. F of the appendix. We measure the loss of PCA dimensions in a local neighborhood to support the diminishing of intrinsic feature dimensions. The results reported in appendix section G and Fig. A8 show a trend of drastic decrease.
> >
> > ### **Q: Does the bottom row of Fig.1 report an exponential trend?**
> >
> > **A:** We have added log scale axes to the bottom row of Fig.1. We can find that they are nearly linear under the log scale axes, which suggests nearly exponential trends of them. We have also rephrased the caption of Fig.1 to turn down the original tone.
> >
> > ### **Q: Figure 1 could also use different colours, especially for the bottom row where distinguishing between Jacobian and Feature rank/dimension is not easy.**
> >
> > **A:** Thank you for raising this point! We have changed the colors used in the bottom row of Fig.1 to make it easier to distinguish between the Jacobi rank and feature dimension.

---

> > > ### Author Response · Authors · 2022-08-02
> > > **Response to Reviewer Dh79: Updates of the revision (Part III)**
> > >
> > > ### **Q: Rephrase the work more in line with Theorem 4 and 5 & other updates as suggested.**
> > >
> > > **A:** We are grateful for the reviewer's valuable suggestions on writing and paper origination. We have substantially revised the manuscript as suggested. We summarize the main updates as follows.
> > >
> > > 1) We shorten sections 3.2 and 3.3, moving the too technical parts to the appendix.
> > >
> > > 2) We split section 4 into three parts.
> > >
> > >     i) Section 4.0 and 4.1 are now merged into one section (sec. 4). This section is designed to give a simple principle of rank diminishing for general almost everywhere smooth deep networks. The merged section is more compact and saves more space for the subsequent contents.
> > >
> > >    ii) Section 4.2 is now becoming an independent section (sec. 5), which aims to study the limiting behavior of ranks in infinitely deep networks. We substantially enhance this section. We add a detailed discussion about why we take the assumptions in Theorem 4 and 5 and how this setting is connected with other aspects of deep learning research. We also discuss how the results of these two theorems will influence the feature representations and how techniques like Batch Norm can help stable training.
> > >
> > >    iii) Section 4.3 and 5 are now merged into one section (sec. 6), which aims to validate the results of previous sections. The validation is split into two parts: validating the diminishing rule in each layer and validating the low-rank structural in the terminal layer. The latter is further split into a numerical validation (the PCA experiment on the last layer) and a semantic validation (the independence deficit).
> > >
> > > The revised version has now put much more attention on the limiting behavior in Theorem 4 and 5, making it the most significant part of this paper, and appears more clear about the roles of each section in this paper. We thank the reviewer again for the helpful advice.

---

> > > > ### Comment · Reviewer_Dh79 · 2022-08-03
> > > > **Rebuttal Response**
> > > >
> > > > Thank you to the authors for their thorough response. I also appreciate the review by Reviewer Gfbz and the authors response there. I am satisfied to increase my score to a 6 (weak accept).
> > > >
> > > > I am curious on the effect of bias parameters on the rank of the hidden layer feature spaces. Intuitively the bias parameters learn to model the average activation of the hidden layer. I appreciate that directions of the ambient space which do not contribute to the rank of the feature space will have little variance, but could the bias (by modelling the shift off of $0$) serve to restore some rank to the layer-wise Jacobian as it will also have to account for the shift from the bias?

---

> > > > > ### Author Response · Authors · 2022-08-04
> > > > > **Thank you very much for supporting our work!**
> > > > >
> > > > > The authors are very grateful for the reviewer's support! Thank you again for your valuable efforts in improving this work!
> > > > >
> > > > > For the role of the bias, yes, it will restore some ranks in many deep networks. We can understand its role from the below two different perspectives.
> > > > >
> > > > > **1. The role of bias in blocks like attention networks**
> > > > > For the networks involving quadric terms like $\sigma_1(W_1x+b_1)^T\sigma_2(W_2x+b_2)$, it is clear that the bias will influence the rank of networks. In the simplest form, $f(x)=[(x+b_1)^T (x+b_2), (x+b_3)^T (x+b_4) ]^T$, then $J_f=[2x+b_1+b_2,2x+b_3+b_4]^T$. Then the rank of $J_f$ is decided by the bias vectors. An example is the attention network, where $s_{ij}=q^T_iv_j, q_i=\alpha_i(A_ix+b_i), v_j=\beta_j(C_jx+d_j)$.
> > > > >
> > > > > In this case, the bias vector influences the rank as they are in fact also the weight vectors; there will be multiplied by some hidden feature representations in the whole network.
> > > > >
> > > > >
> > > > > **2. The role of bias in nonlinearities**
> > > > > For purely linear networks, theoretically, bias will not influence the rank of the network. In fact, the bias term will not influence the Jacobi matrix at all. For example, in $f(x)=W_1(W_2x+b_2)+b_1$, we have $J_f=W_1W_2$ that is independent with the bias vectors. However, in non-linear networks, the nonlinear activations will make the bias responsible for ranks. Consider $f(x)=ReLU(x+b)$, then $J_f$=diag{$\delta_{x_1+b_1>0},\cdots,\delta_{x_n+b_n>0}$}, which is a diagonal square matrix that the $i$-th row and $i$-th column of it is 1 if $x_i+b_i>0$ and is 0 otherwise. Then clearly, the bias vector $b$ will influence the value of the Jacobian matrix and also influence its rank. If $b_i<-sup_{x\in\mathcal{X}} x_i$, then $J_f=0$ and $rank(f)=0$. If $b_i>-\inf_{x\in\mathcal{X}}x_i$, then $J_f$ will  be always of full rank. The bias vector decides how much information of the input manifold $\mathcal{X}$ can be preserved in this layer, thus influencing the rank of the network.

---

### Official Review · Reviewer_Gfbz · 2022-07-09

**Rating:** 5
**Confidence:** 4
**Soundness:** 2 fair
**Presentation:** 3 good
**Contribution:** 3 good

**Summary:**

This paper demonstrates a rank diminishing behavior of deep neural networks considering the mapping from the input space to the  feature space of an increasingly deeper intermedate layer. Theoretically, it proves that the rank doesnot increase as the layer depth increases. Experimentally, it demonstrates a general decreasing trend of rank on various NN architectures. This work also empirically demonstrates that the number of major PCA components at the final feature layer is much less than its ambient dimension, which leads to feak correlation between very different categories.

**Questions:**

1. [related to weakness 2] What exactly is the "rank of its Jacobi matrix Jf over its input domain X" in definition of the rank of function? How the rank of function is related to the rank of its Jacobian matrix at a specific point x\in R^n?
2. [related to weakness 1] Is there any theoretical explanation about why independence deficit happens for the final feature manifold?

**Limitations:**

The authors adequately addressed the limitations.

**Strengths And Weaknesses:**

Strenghs:
1. This work systematically studies the evolution of function rank throughout the layer computation and provides theoretical jusfication to the empirically observed rank diminishing behavior.
2. The finding about the independence deficit of final feature manifolds is very interesting and provides insight to the lack of robustness of DNNs.

Weakness:
1. Classification dimension estimated by the number of major PCA  components in this work is not a good indicator of the feature dimesnion. In fact, a very low dimensional manifold can have high classification dimension. Therefore, the main results about rank diminishing cannot explain the interesting finding about low classification dimension of final feature manifolds. The statement in the abstract that "independence deficit caused by the rank deficiency of deep networks" is misleading.
2. It seems that the definition of the rank of function and lemma 1 implicitly assume that the jacobian of neural network functions has a constant rank over the entire input space of R^n. This is a strong assumption that doesnot hold in general.  When this assumption holds for neural networks should be carefully discussed.

---

> ### Author Response · Authors · 2022-07-28
> **Thank you for the questions and feedback!**
>
> Thank you for the thoughtful feedback and valuable questions! We are encouraged by your comments on the finding of independence deficit! Below we address the questions and concerns separately.
>
>
> ### **Q: The PCA Dimension is not accurate for the feature dimension, and it cannot explain the independence deficit**
>
> **A:**  PCA is indeed not an accurate measure for the intrinsic dimension. But here, we use the PCA dimension based on the following considerations.
>
> **a) PCA dimension offers an upper bound for the intrinsic dimension:** Indeed, a low dimensional feature manifold can have a high PCA dimension (number of significant PCA components). In this case, the PCA dimension gives an *upper bound* for the feature manifold dimension. If this upper bound is low, then the true feature manifold dimension should be even lower. In some rare cases, a high dimensional feature manifold can also have a low PCA dimension, but a small enough tolerance $\epsilon$ for the significant PCA components and large enough sample numbers N can rule out these cases for manifolds having interior points. So when $\epsilon$ is small enough, a low PCA dimension will indicate an equal or even lower intrinsic feature dimension; this will also suggest the independence deficit, which we will explain together with the theory analysis for the source of independence deficit later.
>
>
>  **b) In this paper, the task PCA takes on is qualitatively and not sensitive to accuracy:** In this paper, we only use PCA to qualitatively detect whether the intrinsic dimensions of the terminal feature manifolds are very low. The goal of the ClsDim metric is to support that the rank does diminish a lot so that only a very low dimensional final feature manifold remains. This task is not very sensitive to the accuracy of PCA dimensions. The task of supporting rank diminishing accurately is mainly accomplished by the Partial Rank metric in Fig. 1.
>
> **c) It is hard to find a method that is the same widely recognized but better than PCA:** Overall, we have to admit that measuring the intrinsic dimension of manifolds is a very difficult task, and PCA is one of the most commonly used methods for this task. Under good regularity conditions, the PCA dimension and the intrinsic dimension are connected to each other. If assuming that the feature distribution follows the Gaussian distribution $N(\mu, \Sigma)$, then the PCA dimension, which estimates the dimension of covariance matrix $\Sigma$, can also estimate the feature dimension (which equals the dimension of $\Sigma$). Thus PCA can be viewed as a rough estimation of the intrinsic dimension of feature manifolds.

---

> > ### Author Response · Authors · 2022-07-31
> > **Q: How do we define the rank of function, and why do we assume constant rank in Lemma 1**
> >
> > ### **Q: How do we define the rank of function, and why do we assume constant rank in Lemma 1**
> >
> > **A:** We apologize for the unclearness here.  The constant rank assumption is made for the convenience of expressing the diminishing of feature manifold dimensions and pointing out the connection between network ranks and feature manifolds. For the remained part of the theory analysis, the function rank can be either constant or non-constant (i.e. point-wise), and the conclusions will always stand.
> >
> > **a) Why do we omit the discussion of different ranks:** By our original definition, the rank of function is a pointwise function of the input point $x$. But we can omit the region of non-highest ranks and consider the rank as a constant. The key to omitting the tedious discussion of different ranks in the input domain is [Sard's Theorem](https://en.wikipedia.org/wiki/Sard%27s_theorem) (see the 4-th paragraph of section 'Variants' in this wiki page) we mentioned in Line 67-69. Sard's Theorem tells that **for smooth functions between manifolds, the critical points will be mapped to a zero measure set and thus can be ignored**, where "critical points" means points that have lower ranks than the region of the highest rank. So in a deep neural network, we do not need to consider the region of non-highest ranks, as those regions "contribute zero" to the feature manifolds; they will be mapped to a low-dimensional sub-manifold in the feature manifolds and thus can be ignored when analyzing intrinsic dimensions of the feature manifolds. Thus we only consider the region of the highest rank in Lemma 1, which admits a constant rank $r$ and is also the highest rank that a function can reach in its input domain.
> >
> > **b) The main results of the subsequent theorems hold for the pointwise definition of function ranks:** While we only consider the region of the highest ranks, the results in Theorem 2,4,5 (except Eq. 8) are also applicable for any (differentiable) point in the input domain by changing the notation $rank(F)$ into $rank(F(x))$ and $J_F$ into $J_{F(x)}$. This is because those results are deduced from the chain rule of differential, which holds point-wisely at each differential point $x$ of the input domain. In the experiments, we find that the variance of ranks measured in different points is not significant (as is shown in the error bars of Fig. 1), and the diminishing rule maintains when considering the highest ranks, mean ranks, and the lowerest ranks of the networks.
> >
> > **We have revised the context in sec. 2 to mention the above discussion in the new version of the manuscript. Thank you for raising this point!**

---

> > > ### Author Response · Authors · 2022-07-31
> > > **Q: Is there any theoretical explanation about why independence deficit happens for the final feature manifold?**
> > >
> > > ### **Q: Is there any theoretical explanation about why independence deficit happens for the final feature manifold?**
> > >
> > > **A:** Yes, we can have a much deeper theoretical understanding of this based on the property of LASSO regression. For short, it happens when the axis for a category is close to the orthogonal complementary subspace of the low (numerical) rank covariance matrix of the final output space.
> > >
> > > **Review the problem and its meaning:** Let's review Eq. 11, where we solve the dependence coefficients. For simplicity, we rewrite it as
> > >
> > > $\lambda^*=\min_{\lambda_i=-1}E_{x\sim P_{data}}[\Vert\lambda^Tf(x)\Vert_2^2]+\eta\Vert\lambda\Vert_1=E_{x\sim P_{data}}[\lambda^Tf(x)f(x)^T\lambda]+\eta\Vert\lambda\Vert_1$
> > >
> > > $=\lambda^T E_{x\sim P_{data}}[f(x)f(x)^T]\lambda+\eta\Vert\lambda\Vert_1=\lambda^T\mu\lambda+\eta\Vert\lambda\Vert_1,$
> > >
> > > where $f(x)=WF(x)$ is the slice from the input to the final logits layer of the network, $\mu=E_{x\sim P_{data}}[f(x)f(x)^T]$.  If assume $E[f(x)]=0$, then $\mu$ is the covariance matrix of the logits output.
> > >
> > > For this problem, we have the following observations:
> > > 1) The first term of this objective $E_{x\sim P_{data}}[\Vert\lambda^Tf(x)\Vert_2^2]=E_{x\sim P_{data}}[(\sum_{j\neq i}\lambda_j f_j(x) -f_i(x))^2]$ measures the error of using the linear composition $\sum_{j\neq i}\lambda_j f_j(x)$ to predict the logits for the i-th term $f_i(x)$.
> > > 2) By property of $\ell_1$ penalty, the second term $\eta\Vert\lambda\Vert_1$ enforces sparsity to the coffecitents $\lambda_j, j\neq i$ thus most of $\lambda_j$ will be zero.
> > >
> > >  So the independence deficit (a small cluster of other categories can predict the output for the i category) will happen if and only if 1) $E_{x\sim P_{data}}[\Vert\lambda^{\*T}f(x)\Vert_2^2]=\lambda^{\* T}\mu\lambda^{\*}$ is approaching zero ($f_i(x)\approx \sum_{j\neq i}\lambda_j^{\*} f_j(x),\forall x$), when 2) $\eta$ is relatively large so that most $\lambda_j^{\*}$ is zero.
> > >
> > > **Handling the constraint and $\ell_1$ penalty:** Note that the value of $\lambda^*$ is constraint by two factors in the LASSO problem
> > > 1) The i-th element of it $\lambda_i^{\*}$ has to be -1.
> > > 2) The norm of $\lambda^*$ should be small as it is constrained by the regularization term $\Vert\lambda^*\Vert_1$.
> > >
> > > Combining this two observations, $\lambda^*$ should be a vector lying in the hyper-plane $P_i$={$\lambda\in\mathbb{R}^{1000}:\lambda_i=-1$} and close to the point $(0,...,-1,...,0)$ (the origin point of $P_i$). So, the direction of $\lambda^*$, $\frac{\lambda^*}{\Vert\lambda^* \Vert_2}$ has to be close to the i-th coordinate axis, and $\Vert\lambda^* \Vert_2\geq 1$ as $\lambda_i^{\*}=-1$.
> > >
> > > **When can we reach the first requirement ($\lambda^{\* T}\mu\lambda^{\*}\approx0$):** For a vector with a constraint on norm ($\Vert\lambda^* \Vert_2\geq 1$), it is well known that the value of $E_{x\sim P_{data}}[\Vert\lambda^{\*T}f(x)\Vert_2^2]=\lambda^{\* T}\mu\lambda^{\*}$ will be small if and only if $\lambda^*$ is close to the linear subspace spanned by the singular vectors of $\sqrt{\mu}$ (where $\sqrt{\mu}^T\sqrt{\mu}=\mu$) that correspond to tiny singular values. In this case, $\lambda^{\* T}\mu\lambda^{\*}$ is tiny, and the value of it approaches $\sum_{\sigma_i<<1}\sigma_i^2 (q^T_i\lambda^*)^2\sim C\sum_{\sigma_i<<1}\sigma_i^2$, where $q_i$ is the singular vector of $\sigma_i$ (counting repetitions).
> > >
> > > **Conclusions:** So we can finally conclude that if the i-th coordinate axis is close to the linear subspace spanned by the eigenvectors of $\mu$ that correspond to tiny eigenvalues, then there is a $\lambda^*$ that solves the original problem with a small prediction error $\lambda^{\*T}\mu\lambda^*$ and a small $\ell_1$ norm, which will then induce the independence deficit as we explained in **Review the problem and its meaning**. It then follows that only a low-rank (numerical rank) covariance matrix can have many tiny eigenvalues, and their eigenvectors span a large linear subspace. So the independence deficit will happen only when the covariance of the final outputs is low-rank, and it will happen to the i-th category if the i-th coordinate axis is close to some eigenvectors with small eigenvalues.
> > >
> > > **Further conclusions and connections with low-rank networks:** The above discussion reveals why the independence deficit will happen to a specific category I. If we do not care about which category it happens to, then the low-rank structure of the outputs is the main reason. The lower the numerical rank of the covariance matrix, the larger the space spanned by eigenvectors of tiny eigenvalues, and the higher probability that some of the eigenvectors will be close to some coordinate axis and induce the independence deficit.

---

> > > > ### Author Response · Authors · 2022-08-08
> > > > **We sincerely hope response from the reviewer before the end of the author-reviewer discussion period.**
> > > >
> > > > Dear Reviewer Gfbz,
> > > >
> > > > The author-reviewer discussion period is about to end. Do our responses resolve your initial concerns, and are there any further questions regarding this paper and our new response? We sincerely hope that you can freely share your opinions and suggestions with us and engage with us in the discussion, which is very important for us. We want to thank you again for your time and efforts in improving this work!
> > > >
> > > > We are looking forward to hearing from you.

---

### Meta-Review · Area_Chair_TSa1 · 2022-08-26

**Recommendation:** Accept
**Confidence:** Less certain

**Metareview:**

This paper studied the "rank" of neural networks and showed that deeper network in general will have lower rank. The paper did a detailed empirical study on network rank, as well as some theoretical insights on why rank is likely to decrease as the network becomes deeper, and how the rank decrease can change with or without normalization layers. The paper also demonstrated a "independence deficit" phenomenon which happens when the rank of the output layer is too low. Overall the reviewers feel that the paper gives interesting observations and nice intuitive explanations.

**Award:**

No

---

### Decision · Program_Chairs · 2022-09-14

Accept